# Some Practical Concerns and Solutions for Using Pretrained Representation in Industrial Systems

## Abstract

Deep learning has dramatically changed the way data scientists and engineers craft features – the once tedious process of measuring and constructing can now be achieved by training learnable representations. Recent work shows pretraining can endow representations with relevant signals, and in practice they are often used as feature vectors in downstream models. In real-world production, however, we have encountered key problems that cannot be justified by existing knowledge. They raise concerns that the naive use of pretrained representation as feature vector could lead to unwarranted and suboptimal solution. Our investigation reveals critical insights into the gap of uniform convergence for analyzing pretrained representations, their stochastic nature under gradient descent optimization, what does model convergence means to them, and how they might interact with downstream tasks. Inspired by our analysis, we explore a simple yet powerful approach that can refine pretrained representation in multiple ways, which we call *Featurizing Pretrained Representations*. Our work balances practicality and rigor, and contributes to both applied and theoretical research of representation learning.

## 1 Introduction

The ability of neural networks to learn predictive feature representation from data has always fascinated practitioners and researchers (Bengio et al., 2013). The learnt representations, if proved *reliable*, can potentially renovate the entire life cycle and workflow of industrial machine learning. Behind *reliability* are the three core principles for extracting information from data, namely **stability**, **predictability**, and **computability** (Yu, 2020). These three principles can not only justify the practical value of learnt representation, but also lead to the efficiency, interpretability, and reproducibility that are cherished in real-world production. Since pretrained representations are optimized to align with the given task, intuitively, they should satisfy all three principles in a reasonable setting. However, when productionizing an automated pipeline for pretrained representations in an

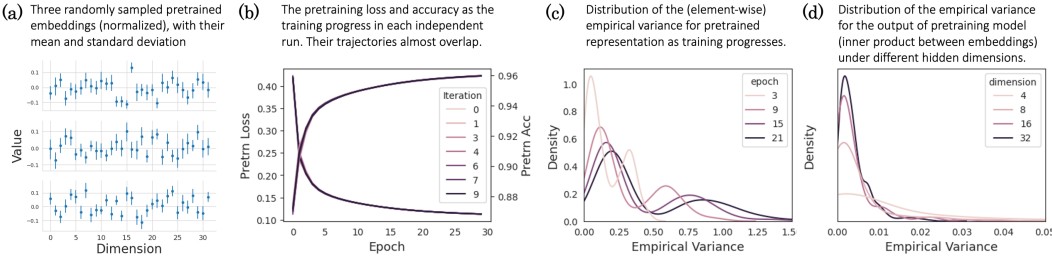

Figure 1: Illustrating the stability issue of pretrained representation with **MovieLens-1m**. The details of the experiments are deferred to Appendix F. The empirical variances are computed from ten independent runs.

industrial system, we encountered key problems that cannot be justified by existing knowledge. In particular, while the daily refresh follows the same modelling and training configurations and uses essentially the same data[1], downstream model owners reported *unexpectedly high fluctuations* in

---

[1]Since the pretraining uses years of history data, the proportion of new daily data is quite small.

performance when retraining their models. For illustration purpose, here we reproduce the issue using benchmark data, and take one further step where the pretraining is *repeated* on exactly the *same data*, under the *same model configuration*, *training setup*, and *stopping criteria*. We implement ten independent runs to essentially generate the $i.i.d$ versions of the pretrained representation.

We first visualize the dimension-wise empirical variances of the pretrained representations, provided in Figure 1a. It is surprising to find out that while the pretraining losses almost converge to the same value in each run (Figure 1b), there is such a high degree of uncertainty about the exact values of each dimension. Further, in Figure 1c, we observe that the uncertainty (empirical variance) of pretrained representation will increase as the pretraining progresses. In the downstream task where pretrained representations are used as feature vectors (see the *right figure*), we observe that the performance does fluctuate wildly from run to run. Since we use logistic regression as the downstream model, the fluctuation can only be caused by the *instability* of pretrained representations because we can effectively optimize the downstream model to global optimum. To demonstrate that the above phenomenon is not caused by using a specific model or data, we also experiment with a completely different pretraining model and benchmark data from from another domain. We perform the same analysis, and unfortunately the same issues persist (Figure A.1 in the Appendix).

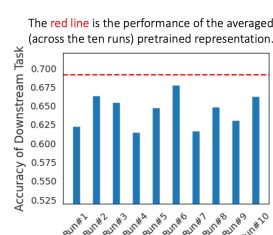

Existing deep learning theory, both the convergence and generalization results (we will discuss them more in Section 2), can fail to explain **why shall we expect pretrained representation to work well in a downstream task when their exact values are so unstable**. This is especially concerning for industrial systems as the issue can lead to unwarranted and suboptimal downstream solutions. We experienced this issue firsthand in production, so we are motivated to crack the mysteries behind pretrained representations, and understand if and how their stability can be improved without sacrificing predictability and computability. We summarize our contributions as below.

- We provide a novel uniform convergence result for pretrained representations, which point out gaps that relate to the *stability* and *predictability* issues.

- We break down and clarify the *stability issue* by revealing the stochastic nature of pretrained representation, the convergence of model output, and the stable and unstable components involved.

- We investigate the interaction between pretrained representation and downstream tasks in both parametric and non-parametric settings, each revealing how *predictability* can benefit or suffer from stability (or instability) for particular usages of pretrained representations.

- We discuss the idea of *featurizing pretrained representation*, and propose a highly practical solution that has nice guarantees and balances *stability*, *predictability*, and *computability*. We also examine its effectiveness in real-world experiments and online testings.

## 2    RELATED WORK

It is not until recent years that deep learning theory sees major progress. Zhang et al. (2016) observed that parameters of neural networks will stay close to initialization during training. At initialization, wide neural networks with random weights and biases are *Gaussian processes*, a phenomena first discussed by Neal (1995) and recently refined by Lee et al. (2017); Yang (2019). However, they do not consider effect of optimization. The *Neural Tangent Kernel* provides a powerful tool to study the limiting convergence and generalization behavior of gradient descent optimization (Jacot et al., 2018; Allen-Zhu et al., 2019), but it sometimes fails to capture meaningful characteristics of practical neural networks (Woodworth et al., 2020; Fort et al., 2020). However, those works require parameters being close to initialization, in which useful representation learning would not take place.

Indeed, it has also caught to people's attention that representation learning can go beyond the neural tangent kernel regime (Yehudai & Shamir, 2019; Wei et al., 2019; Allen-Zhu & Li, 2019; Malach et al., 2021), among which a line of work connects the continuous-time training dynamics with *mean field* approximation (Mei et al., 2018; Sirignano & Spiliopoulos, 2020), and another direction is to study the *lazy training* regime (Chizat et al., 2019; Ghorbani et al., 2019) where only the last layer of a neural network is trained. Unfortunately, their assumed training schemas all deviate from practical

representation learning. Still, part of our analysis in Section 4.2 can be viewed as a practical *discrete-time extension* of the mean-field method. Perhaps the most practical setting for studying pretrained representation is Arora et al. (2019), which analyzes the *contrastive representation learning* under a particular data generating mechanism. However, their results do not generalize to broader setting, and they cannot justify the stability issue of pretrained representation.

## 3 PRELIMINARIES

**Notations**. We use $x \in \mathcal{X} \subseteq \mathbb{R}^{d_0}$ and $y \in \mathbb{R}$ to denote the raw feature and outcome, upper-case letters to denote random variables and measures, and bold-font letters to denote matrices. Let $h : \mathcal{X} \to \mathbb{R}^d$ be the representation hypothesis, and $f : \mathbb{R}^d \to \mathbb{R}$ be the prediction hypothesis. The hypothesis classes are given by $\mathcal{H}$ and $\mathcal{F}$ respectively. Denote by $\circ$ the operator for function composition, and $\ell : \mathbb{R} \times \mathbb{R} \to [0, 1]$ the loss function. We assume $\ell$ is 1-Lipschitz without loss of generality. Then the risk for a pair of $(h \in \mathcal{H}, f \in \mathcal{F})$ is given by: $R(h, f) := \mathbb{E}_{(X,Y)\sim P}\big[\ell\big(f \circ h(X), Y\big)\big]$, where $P$ is a measure on $(\mathcal{X}, \mathbb{R})$. We also use $P^n$ to denote the corresponding product measure for $(X_1, Y_1), \ldots, (X_n, Y_n)$.

The one-layer *multi-layer perceptron* (*MLP*) is perhaps the most fundamental representation learning model, given by: $f \circ h(x) = \Theta\sigma(\mathbf{W}x)$. Here, $\sigma$ is the activation function, and $\mathbf{W} \in \mathbb{R}^{d_0 \times d}$, $\Theta \in \mathbb{R}^{d \times k}$. We mention that adding the bias terms will not affect our analysis, so we drop them here for brevity. In practice, $\Theta$ and $\mathbf{W}$ are often initialized as *scaled i.i.d* Gaussian random variables that follow $N(0, 1/d)$. We will use such as $[W]_i$ to denote the $i^{th}$ row of a matrix. The popular contrastive representation learning can also be considered as a special case of this configuration[2].

Define the shorthand $g(x) := \Theta\sigma(\mathbf{W}x)$. A typical pretraining process involves optimizing the risk function defined for pretraining and extracting the hidden representation. The optimization is done via *stochastic gradient descent* (*SGD*), e.g. $\mathbf{W}^{(t+1)} = \mathbf{W}^{(t)} - \alpha\nabla_{\mathbf{W}}\ell(g(x^{(t)}), y^{(t)})$, where $\alpha$ is the learning rate. For convenience, we consider each mini-batch having one random sample, denoted by $(x^{(t)}, y^{(t)})$ that corresponds to the $t^{th}$ step.

Given a representation hypothesis $h$, we define: $f_{h,n} := \arg\min_{f \in \mathcal{F}} 1/n \sum_{i=1}^n \ell\big(f(h(x_i), y_i)\big)$. In the sequel, how well $f_{h,n} \circ h$ can generalize to a new *i.i.d* sample of the downstream task is:

$$R(h) := \mathbb{E}_{(X,Y)\sim P}\mathbb{E}_{P^n}\big[\ell\big(f_{h,n} \circ h(X), Y\big)\big],$$

where the second expectation $\mathbb{E}_{P^n}$ is taken with respect to the downstream data $\{X_i, Y_i\}_{i=1}^n$ underlying $f_{h,n}$. Its empirical version is given by $R_n(h) := 1/n \sum_i \big[\ell\big(f_{h,n} \circ h(X_i), Y_i\big)\big]$.

## 4 MAIN ANALYSIS

### 4.1 THE GAP OF UNIFORM CONVERGENCE FOR PRETRAINED REPRESENTATION

Suppose $h$ and $f$ are optimized jointly (end-to-end) via *empirical risk minimization* (*ERM*), which amounts to solving: $\arg\min_{h \in \mathcal{H}, f \in \mathcal{F}} 1/n \sum_i \ell(f \circ h(x_i), y_i)$. In this setting, the generalization behavior of the solution well-studied. In particular, using the notion of Gaussian (or Rademacher) complexity[3], the generalization error can be bounded by $\mathcal{O}\big(\mathcal{G}_n(\mathcal{F} \circ \mathcal{H})/n + \sqrt{(\log 1/\delta)/n}\big)$ with probability at least $1 - \delta$ (Bartlett & Mendelson, 2002). This result, known as uniform convergence, is especially appealing because it both includes problem-specific aspects and applies to all functions in the composite hypothesis class $\mathcal{F} \circ \mathcal{H} := \{f \circ h : f \in \mathcal{F}, h \in \mathcal{H}\}$.

Is it possible to achieve a comparable result for pretrained representation? Perhaps the most ideal setting for uniform convergence to hold under pretrained representation is:

**C1:** the pretraining and downstream training will use the same data $\{(X_i, Y_i)\}_{i=1}^n$, i.e. $\hat{h}, \hat{f} := \arg\min_{h \in \mathcal{H}, f \in \mathcal{F}} \frac{1}{n} \sum_{i=1}^n \ell\big(f \circ h(X_i), Y_i\big)$, $f_{\hat{h},n} = \arg\min_{f \in \mathcal{F}} \frac{1}{n} \sum_{i=1}^n \ell\big(f(\hat{h}(X_i), Y_i)\big)$.

---

[2]We can simply set $x_i \in \mathbb{R}^n$ as one-hot encodings, and $\mathbf{W}, \Theta \in \mathbb{R}^{d_0, d}$ where they are allowed to coincide. Then we let $h(x_i) = [\mathbf{W}]_i$ or $[\Theta]_i$ depending on the context. The activation becomes the identity function, and $\ell(f(x_i), x_j) = \log(1 - \sigma(h(x_i)^T h(x_j)))$ (or $\log \sigma(h(x_i)^T h(x_j))$, with $\sigma(\cdot)$ being the Sigmoid function.

[3]We will use Gaussian complexity $\mathcal{G}(\cdot)$ here for some of its technical convenience. Then we let $\mathcal{G}_n$ be the empirical Gaussian complexity. See Appendix A for detail.

**C2:** they rely on the same prediction function class $\mathcal{F}$.

These two conditions essentially eliminate the confounding effects of model and data mismatch. Thus, if uniform convergence cannot hold in this setting, it is unlikely to serve more general use cases. We first summarize the common intuition behind why pretrained representation might work:

- the pretraining objective, when well-designed, reasonably predicts the empirical downstream risk for $f_{h,n}$ (**intuition 1**);
- $f_{h,n}$'s empirical downstream risk can be generalized to the true downstream risk (**intuition 2**).

These two intuitions have also been exemplified for contrastive representation learning in Arora et al. (2019) and its following work. Our main contribution here is to make the above intuitions rigorous, and reveal whether they are indeed sufficient for uniform convergence in general settings.

Recall that, given the complete information on a downstream task, the best we can do is: $\min_{h \in \mathcal{H}, f \in \mathcal{F}} R(h, f)$. We denote the representation hypothesis that achieves this minimum by $h^*$. Let $\hat{h}$ be given in **C1**. Then the generalization error is simply given by: $R(\hat{h}) - \min_{h \in \mathcal{H}, f \in \mathcal{F}} R(h, f)$. Following the standard derivation which decomposes the generalization error and takes the supremum to upper bound each term, we run into terms that exactly characterize the above two intuitions. As we show in Appendix B, it holds that:

$$R(\hat{h}) - \min_{h \in \mathcal{H}, f \in \mathcal{F}} R(h, f)$$

$$\leq \sup_h \left\{ \mathbb{E}_{P^n} R_n(h) - R_n(h) \right\} + \sup_h \mathbb{E}_{P^n} \left[ \mathbb{E}_{(X,Y) \sim P} \left[ \ell\big(f_{h,n} \circ h(X), Y\big) - R_n(h) \right] \right] + \text{remainder},$$

where the first term $\sup_h \left\{ \mathbb{E}_{P^n} R_n(h) - R_n(h) \right\}$ exactly seeks to match **intuition 1**, and the second term can be further upper bounded using: $\mathbb{E}_{(X,Y) \sim P} \left[ \ell\big(f_{h,n} \circ h(X), Y\big) - R_n(h) \right] \leq \sup_f \left\{ \mathbb{E}_{(X,Y) \sim P} \left[ \ell\big(f \circ h(X), Y\big) - R_n(h) \right] \right\}$, which underlies **intuition 2**. The remainder terms can be bounded using standard concentration results.

However, we also spot a critical issue with the first term, and we first expand it for clarity:

$$\sup_h \left\{ \mathbb{E}_{P^n} \left[ \frac{1}{n} \sum_i \ell\big(f_{h,n} \circ h(X_i), Y_i\big) \right] - \frac{1}{n} \sum_i \ell\big(f_{h,n} \circ h(X_i), Y_i\big) \right\}.$$

Notice that this is not the typical empirical process encountered in a standard generalization setting, and we show that its upper bound is actually given by $\mathcal{O}\big(\mathcal{G}_n(\mathcal{H})/\sqrt{n} + \sqrt{\log 1/\delta}\big)$ following the same procedure as Bartlett & Mendelson (2002). Compared with the standard generalization bound, here the slack term $\sqrt{\log 1/\delta}$ does not vanish as we increase $n$. Therefore, there exist gaps between common intuitions and achieving uniform convergence. Before we discuss the cause of the gaps and its implications, we first present the complete result as below.

**Proposition 1.** *Let $\mathcal{G}'_n(\cdot)$ be a slightly modified Gaussian complexity term. Under the conditions and definitions in **C1** and **C2**, it holds with probability at least $1 - \delta$ that:*

$$R(\hat{h}) - \min_{h \in \mathcal{H}, f \in \mathcal{F}} R(h, f) \lesssim \frac{\mathcal{G}_n(\mathcal{H})}{\sqrt{n}} + \frac{\mathcal{G}'_n(\mathcal{F}) \sup_h \sqrt{\mathbb{E}\|h(X)\|_2^2}}{\sqrt{n}} + \sqrt{\log 1/\delta}.$$

The proof is deferred to Appendix B. Proposition 1 can be viewed as a "*no free lunch*" result for using pretrained representation: even in the most ideal setting we study here, uniform convergence cannot be expected for all representation hypothesis. The gap is that not for every $h \in \mathcal{H}$ can the pretraining objective be predictive of $f_{h,n}$'s empirical downstream risk. Imagine a entirely random $\tilde{h}$. Then both its pretraining objective and the empirical downstream risk of $f_{\tilde{h},n}$ may have high variances that do not scale with $n$. Thus the prediction will not concentrate whatsoever.

**Takeaway.** The implications of this gap are two folds. Firstly, it does not suffice to only study $\mathcal{H}$ and the data distribution – the ***statistical*** and ***algorithmic*** convergence properties of $\hat{h}(X)$ could be more relevant as they suggest its ***stability***. Secondly, we cannot take the performance of $f_{\hat{h},n}$ for granted, at least not without understanding how $\hat{h}(X)$ interacts with the downstream learning and generates $f_{\hat{h},n}$ – which ultimately relates to its ***predictability***. Unfortunately, we find a lack of discussion on these two issues in existing literature, so we will investigate them in the next two sections.

### 4.2 STOCHASTIC NATURE OF PRETRAINED REPRESENTATION, AND THE CONVERGENCE OF PRETRAINING MODEL

In this section, we reveal two important **statistical** and **algorithmic** properties of pretrained representation. We show that while they persist as random vectors during SGD optimization (as shown in Figure 1), the output of the pretraining model can be deterministic and converge to some optimal solution. Two contributing factors are *scaled i.i.d initialization* and the *inductive bias of gradient descent*. Our findings provide critical insight to the stability of pretrained representations.

We motivate our statistical analysis by deriving the optimization path of the one-layer MLP introduced in Section 3. For notation convenience, we introduce $\tilde{\mathbf{\Theta}}$ and $\tilde{\mathbf{W}}$ as the *rescaled* version of $\mathbf{\Theta}$ and $\mathbf{W}$ such that $\tilde{\mathbf{\Theta}}^{(0)}, \tilde{\mathbf{W}}^{(0)} \overset{i.i.d}{\sim} N(0,1)$. We let $\ell'(g(x), y)$ be the derivative of the loss function and similarly for other functions. In contrast to the existing theoretical work that studies optimization path under gradient flow or infinitesimal learning rate, we fix the learning rate as $\alpha = 1$ to reflect real-world practice. The output dimension is also set to $k = 1$ without loss of generality.

In the first forward pass, since $\sigma(\mathbf{W}^{(0)} x^{(0)})$ has $i.i.d$ coordinates, as $d \to \infty$ it holds that:

$$g^{(0)}(x^{(0)}) := \frac{1}{d} \sum_{i=1}^{d} \big[\tilde{\mathbf{\Theta}}^{(0)}\big]_i \big[\sigma\big(\tilde{\mathbf{W}}^{(0)} x^{(0)}\big)\big]_i \overset{a.s.}{\longrightarrow} \mathbb{E}\Theta^{(0)} \sigma\big(W^{(0)} x^{(0)}\big) \quad \text{(denote by } g_*^{(0)}(x^{(0)})),$$

where we use $\Theta^{(t)}, W^{(t)}$ to denote an $i.i.d$ element (or row) of $\tilde{\mathbf{\Theta}}^{(t)}$ and $\tilde{\mathbf{W}}^{(t)}$. As a result, $\ell'\big(g^{(0)}(x^{(0)}), y^{(0)}\big)$ also converges to the deterministic value $\mathcal{L}^{(0)} := \ell'\big(g_*^{(0)}(x^{(0)}), y^{(0)}\big)$. Then in the first backward pass, the updated parameters will follow:

$$\tilde{\mathbf{\Theta}}^{(1)} = \tilde{\mathbf{\Theta}}^{(0)} - \mathcal{L}^{(0)} \sigma\big(\tilde{\mathbf{W}}^{(0)} x^{(0)}\big), \quad \tilde{\mathbf{W}}^{(1)} = \tilde{\mathbf{W}}^{(0)} - \mathcal{L}^{(0)} x^{(0)} \tilde{\mathbf{\Theta}}^{(0)} \sigma'\big(\tilde{\mathbf{W}}^{(0)} x^{(0)}\big).$$

An important observation is that the updated parameters remain to be element-wise $i.i.d$. Consequently, the model output of the second forward pass will also converge to a deterministic value:

$$g^{(1)}(x^{(1)}) \overset{a.s.}{\longrightarrow} \mathbb{E}\Big(\Theta^{(0)} - \mathcal{L}^{(0)} \sigma\big(W^{(0)} x^{(1)}\big)\Big)\Big(W^{(0)} x^{(1)} - \mathcal{L}^{(0)} x^{(0)} \Theta^{(0)} \sigma'\big(W^{(0)} x^{(0)}\big) x^{(1)}\Big).$$

As we show in the following Proposition, the (statistical) convergence result will hold for any $t$, and there exists a general iterative update rule for $g^{(t)}(x)$. For some intuition, suppose $\sigma(\cdot)$ is the identity function, then $\mathbf{\Theta}^{(t)}, \mathbf{W}^{(t)}$ will simply be linear combinations of $\mathbf{\Theta}^{(0)}, \mathbf{W}^{(0)}$.

**Proposition 2.** *For the one-layer MLP we consider, with the learning rate $\alpha = 1$, for any step $t > 1$, as $d \to \infty$, the model output $g^{(t)}(x)$ will converge almost surely to $g_*^{(t)}(x)$ defined as follows:*

$$g_*^{(t)}(x) = \big(C_1^{(t)} C_2^{(t)} + C_3^{(t)} C_4^{(t)}\big) x,$$

*with $\big(C_1^{(t+1)}, C_2^{(t+1)}, C_3^{(t+1)}, C_4^{(t+1)}\big) = \big(C_1^{(t)}, C_2^{(t)}, C_3^{(t)}, C_4^{(t)}\big) + \mathcal{L}^{(t)} x^{(t)} \big(C_3^{(t)}, C_4^{(t)}, C_1^{(t)}, C_2^{(t)}\big).$*

As a corollary, while the hidden representations will remain random vectors throughout the SGD process (which can be seem from the update rule):

$$h^{(t)}(x) := \sigma(\mathbf{W}^{(t)} x) = \sigma\Big(\tilde{\mathbf{W}}^{(t-1)} x - \mathcal{L}^{(t-1)} x^{(t-1)} \tilde{\mathbf{\Theta}}^{(t-1)} \sigma'\big(\tilde{\mathbf{W}}^{(t-1)} x^{(t-1)}\big) x\Big),$$

$\langle h^{(t)}(x), h^{(t)}(x') \rangle$ will nevertheless also converge to some deterministic value as $d \to \infty$. The proof and detail are deferred to Appendix C. In Figure 1d, we see that the statistical convergence of model output is indeed evident even with moderately small $d$, and its variance is by magnitudes smaller than the variance of the hidden representation $\sigma(\mathbf{W}^{(t)} x)$ (see the x-axis of Figure 1c and 1d).

On the other hand, the algorithmic convergence of model prediction has received considerable attention. It has been shown that over-parameterized models will converge to *minimum-norm interpolants* due to the inductive bias of gradient descent (Bartlett et al., 2021; Soudry et al., 2018). For the sake of space, here we focus on their implications and leave the details to Appendix C. Roughly speaking, among the many locally optimum solutions that interpolate the training data, gradient descent will **converge to the one with the smallest norm**, which usually has nice properties such as smoothness. We let $g_0$ be that particular solution such that $\lim_{t \to \infty} g^{(t)}(x) = g_0(x)$. Since $\langle h^{(t)}, h^{(t)} \rangle$ converge statistically to a deterministic value at every optimization step, we can immediately conclude that:

- if $g^{(t)}$ takes the form of $\langle h^{(t)}, h^{(t)} \rangle$ such as in contrastive representation learning, the inner product between hidden representations also converge algorithmically to $g_0$'s prediction;

- if $g^{(t)} = \theta h^{(t)}$, i.e. the last hidden layer is used as the representation, note that a necessary but not sufficient condition for $\|g^{(t)}(x) - g^{(t)}(x')\|$ to be small is that $\|h^{(t)}(x) - h^{(t)}(x')\|$ is small as well. Suppose $h^{(t)}$ are normalized, then upon the algorithmic convergence, $\langle h^{(t)}(x), h^{(t)}(x') \rangle$ are likely to be larger if $x, x'$ are close to each other under $g_0$'s prediction.

**Takeaway**. The stochastic nature of $\hat{h} := \lim_{t \to \infty} h^{(t)}$ and the (approximate) convergence of $\langle \hat{h}(x), \hat{h}(x') \rangle$ under gradient descent reveal two important properties of pretrained representations:

1. **Instability of $\hat{h}(x)$**: the exact position of $\hat{h}(x)$ in $\mathbb{R}^d$ is stochastic, depending on the initialization and the order of the pretraining data that is fed to SGD;

2. **Stability of $\langle \hat{h}(x), \hat{h}(x') \rangle$**: the pairwise inner product of $\langle \hat{h}(x), \hat{h}(x') \rangle$ converges (approximately) to a value that is consistent with the minimum-norm interpolant of the pretraining task.

These results will also play a crucial role in understanding how $\hat{h}$ can interact with the downstream learning, which we will study in the next section.

## 4.3 INTERACTION WITH DOWNSTREAM TASK

To be comprehensive, we consider both the parametric and non-parametric set up for downstream task. Interestingly, they will reveal different aspects on the ***predictability*** of $\hat{h}$.

**Parametric setup**. To eliminate the interference of label noise, we consider the noiseless setting where the output of downstream task is generated by: $y_i = f^*\big(\mathbb{E}[h(x_i)]\big), i = 1, \ldots, n$. Because $h(x)$ might be high-dimensional, we assume there is some sparsity in $f^*$. The conditions below provide perhaps the easiest parametric setup for pretrained representations to perform well.

**C3:** Let $f^*(h) := \langle \theta^*, h \rangle$, $\|\theta^*\|_0 \leq q$, and let the inputs $h_i := \mathbb{E}h(x_i)$ be sampled from: $N(0, \sigma_h^2 \mathbf{I})$ where $\sigma_h$ is the strength of the signal. We show previously that $\hat{h}$ is stochastic, so we simply set $\hat{h}_i := h_i + \epsilon_i$, where $\epsilon_i \sim N(0, \sigma_\epsilon^2 \mathbf{I})$ captures the variance of the pretrained representation.

Intuitively, since $\epsilon_i$ are $i.i.d$, it holds that $\mathbb{E}_\epsilon\big[\langle \hat{h}(x_i), \hat{h}(x_j) \rangle\big] = \langle h(x_i), h(x_j) \rangle$ so recovering $\theta^*$ should be less challenging. However, we show that the variance will again prohibit efficient learning, and the best $f_{\hat{h},n}$ can do is controlled by $\sigma_\epsilon/\sigma_h$ – a notion of *signal-to-noise ratio* for pretrained representation. The result below takes the form of a *minimax lower bound*: an information-theoretical quantity that characterize the inherent difficulty of a problem. Our proof (in Appendix D) is based on *Le Cam's method* that was previously used to prove a lower bound result under label noise (Raskutti et al., 2011), which is very different from our setting.

**Proposition 3.** *Under **C3**, it holds with probability at least $1/2$ that:*

$$\inf_{\hat{\theta}} \sup_{\|\theta^*\|_0 \leq q} \|\hat{\theta} - \theta^*\|_2 \gtrsim \big(\sigma_\epsilon^2/\sigma_h^2\big) \cdot q n^{-1} \log(d/q),$$

*where $\inf_{\hat{\theta}}$ is taken with respect to any learning procedure that is based on $\{\hat{h}(x_i), y_i\}_{i=1}^n$.*

**Takeaway**. The result in Proposition 3 is alarming because during pretraining, the variance of $h(x)$ might increase as more and more stochastic terms are being added (suggested by both the derivations in Section 4.2 and the empirical result in Figure 1c). The above lower bound shows the predictability of $\hat{h}(x)$ can be compromised by its variance inherited from pretraining. This also explains the instability in downstream machine learning that we experienced during real-world production.

**Non-parametric setup**. Among the non-parametric regression estimators, the *Nadaraya-Watson* (NW) estimator has received considerable attention due to its simplicity and effectiveness (Nadaraya, 1964). It can be thought of as a smoothing nearest-neighbor estimator under a weighting schema:

$$f_{h,n} \circ h(x) := \sum_{i=1}^n y_j w_h(x, x_i), \quad w_h(x, x_i) := K\big((h(x) - h(x_i))/z\big),$$

where $K : \mathbb{R}^d \to \mathbb{R}^+$ is a kernel, and $z$ is a normalizing constant. Here, we omit the bandwidth parameter for convenience. The *Gaussian kernel* $K(u) \propto \exp(-\|u\|_2^2)$ is a common choice, so when pretrained representations are normalized, it only depends on $h$ via $\langle h(x), h(x') \rangle$ – a more *stable* quantity according to the previous section. We effectively denote this kernel by $K(\langle h(x), h(x') \rangle)$. It is well-understood that the generalization of a *kernel support vector machine* is controlled by the *kernel-target alignment* (Cristianini et al., 2001), i.e. $\langle \vec{y}, \mathbf{K}\vec{y} \rangle$, where $\vec{y} = [y_i, \ldots, y_m]^T$ and $\mathbf{K}_{i,j} = K(\langle h(x_i), h(x_j) \rangle)$. We prove that this is also the case for NW estimator, with a simple result that does not resort to the concentration arguments. The proof is in Appendix D.

**Lemma 1.** *Under 0-1 loss, with probability at least $1 - \delta$, the risk of NW estimator satisfies:*

$$R(f_{h,n} \circ h) \leq 1 - \sqrt{\delta} \cdot \mathbb{E}\Big[ \mathbb{1}[Y = Y']K(\langle h(X), h(X') \rangle) \Big],$$

*where the expectation is taken with respect to $(X, Y) \sim P$, $(X', Y') \sim P$.*

**Takeaway**. Lemma 1 shows the *predictability* of $h(x)$, when expressed and measured through the more stable $\langle h(x), h(x') \rangle$, is strictly guaranteed. Therefore, using $h(x)$ in downstream task in the form of $\vec{h}(x) := \big[ e^{\langle h(x), h(x_1) \rangle}, \ldots, e^{\langle h(x), h(x_n) \rangle} \big]$ can be beneficial, and it can be interpreted as a *representation of weights* in the NW estimator. Further, $\vec{h}(x)$ contains all the pairwise relationship that can be more closely related to the pretraining objective. Note that $h(x)$ can also be viewed as the compression of $\vec{h}(x)$ because: $[\vec{h}(x_i)]_j = \exp(\langle h(x_i), h(x_j) \rangle)$. Nevertheless, $\vec{h}(x)$ and $h(x)$ cannot be compared directly because they have different intrinsic dimensionality. In terms of *computability*, $\vec{h}(x) \in \mathbb{R}^n$ is also no compare to $h(x) \in \mathbb{R}^d$ – computing $\vec{h}(x)$ itself can be non-trivial for large-scale applications. We aim to resolve these issues in the next section.

## 5 FEATURIZING PRETRAINED REPRESENTATION

Our next goal is to build on top of $h(x)$ features or representations that are comparable to $\vec{h}(x)$ in terms of *stability* and *predictability*, and have similar *computability* to $h(x)$.

Suppose $\{h(x_i)\}_{i=1}^n$ are normalized. Then $\vec{h}(x_i)$ is simply the exponential of pairwise cosine distances between $h(x_i)$ and all the pretrained representations. Notice that the angle between any pair of $(h(x_i), h(x_j))$ can be decomposed into their respective angle with a *baseline direction* $u \in \mathbb{R}^d, \|u\|_2 = 1$. When the set of baseline directions is rich enough, we can recover all the pairwise cosine distances in $\vec{h}(x_i)$ using their angles with the baseline directions. Given $U := [u_1, \ldots, u_m] \in \mathbb{R}^{d \times m}$, the set of angles between $h(x_i)$ and $U$ forms a *measurement* for the relative location of $h(x) \in \mathbb{R}^d$. We refer to such a measurement process as *featurizing pretrained representation*, as it is similar to how features are constructed by measuring experimental subjects.

While featurizing $h(x)$ according to its geometrically property is an appealing solution, it is unknown how many baseline directions are needed to preserve the *stability* and *predictability* of $\vec{h}$, as well as the optimal way to choose those directions. Fortunately, the *Bochner's Theorem* (Loomis, 2013) from harmonic analysis lays a solid foundation for selecting the directions and providing approximation and learning guarantees. Also, the resulting measurements will coincide with the *random Fourier feature* (Rahimi & Recht, 2007; Liu et al., 2021) that plays a critical role in many machine learning communities. For the Gaussian kernel we studied, Bochner's Theorem states that there exists a measure $Q$ on $\mathbb{R}^d$ such that:

$$K(h(x), h(x')) = \int_{\mathbb{R}^d} e^{iu(h(x) - h(x'))} q(u) du \stackrel{\text{real part}}{=} \mathbb{E}_{u \sim Q}\Big[ \cos\big(u(h(x_1) - h(x_2))\big) \Big].$$

Since $\cos(a - a') = \cos(a)\cos(a') + \sin(a)\sin(a')$, we can approximate the kernel value using the *Monte Carlo* method as below:

$$K(h(x), h(x')) \approx \frac{1}{m} \sum_{i=1}^m \cos\big(u_i h(x)\big) \cos\big(u_i h(x')\big) + \sin\big(u_i h(x)\big) \sin\big(u_i h(x')\big), \quad u_i \stackrel{i.i.d}{\sim} Q.$$

Let $\phi_m(h(x), Q) := 1/\sqrt{m}\big[ \cos\big(u_1 h(x)\big), \sin\big(u_1 h(x)\big), \ldots, \cos\big(u_m h(x)\big), \sin\big(u_m h(x)\big) \big]$ be the featurization of $h(x)$ according to Bochner's Theorem. Note that it amounts to measuring $h(x)$'s

distances with respect to random directions drawn from $Q(u)$, and then transforming them through trigonometric functions. Furthermore, $\langle \phi_m(h(\cdot), Q), \phi_m(h(\cdot), Q) \rangle$ can approximate any entries in $\vec{h}$. To be more precise, Rahimi & Recht (2007) shows that it only requires $m = \Omega\big(d/\epsilon^2 \log(\sigma_Q/\epsilon)\big)$ to achieve $\big|K(h(x), h(x')) - \langle \phi_m(h(x), Q), \phi_m(h(x'), Q) \rangle\big| \leq \epsilon$, where $\sigma_Q^2$ is the second moment $Q$. Therefore, when $m$ is comparable to $d$, the featurized $\phi_m(h(x), Q)$ achieves the *stability* and *predictability* of $\vec{h}$, as well as the *computability* of $h$.

Converting $h(x)$ to $\phi_m(h(x), Q)$ is computationally efficient, since $u_1, \ldots, u_m$ only need to be drawn from Q once and apply to all $h(x_i), i = 1, \ldots, n$. However, there is still the obstacle of finding the optimal $Q^*$. Strictly speaking, $Q^*$ is obtained from the inverse Fourier transform, but in practice the standard Gaussian distribution is often used. Indeed, compute the inverse Fourier transform and sample from it poses another challenging task. To our knowledge, there is no existing study on whether we can safely sample $u$ from a proxy $Q$. In the following proposition, we show that using $Q$ instead of $Q^*$ *will not cost stability* as long as their discrepancy is bounded.

In particular, we state our result in the context of Lemma 1, that is, the downstream risk is controlled by the *alignment* $A := \mathbb{E}\big[\mathbb{1}[Y = Y']K\big(\langle h(X), h(X') \rangle\big)\big]$. We use $D_s(Q, Q^*) := \int s(dQ/dQ^*)dQ^*$ to denote the *f-divergence* induced by $s(\cdot)$.

**Proposition 4.** *Let* $\mathcal{Q}(Q^*; \delta) := \{Q : D_s(Q, Q^*) \leq \delta\}$ *be a* $D_s$*-ball with radius* $\delta$ *centered at* $Q^*$*. Let* $\{h(x_i), y_i\}_{i=1}^n$ *be the downstream data, and* $A_n(Q) := \frac{1}{n(n-1)} \sum_{i \neq j} \mathbb{1}[y_i = y_j]\langle \phi_m(h(x_i), Q), \phi_m(h(x_i), Q) \rangle$*. It holds that:*

$$Pr\Big( \sup_{Q \in \mathcal{Q}(Q^*; \delta)} \big| A_n(Q) - A_n(Q^*) \big| \geq \epsilon \Big) \lesssim \frac{\sigma_{\mathcal{Q}}^2}{\epsilon^2} \exp\Big( -\frac{m\epsilon^2}{16(d+2)} \Big) + \exp\Big( -\frac{n\epsilon^2}{64(1+\delta)} \Big),$$

*where* $\sigma_{\mathcal{Q}} := \max_{Q \in \mathcal{Q}} \sigma_Q$*.*

The significance of Proposition 4 is that even if the optimal $Q^*$ is not used, in the worst case scenario, the instability caused by it (reflected via $\delta$) vanishes quickly as the sample size gets larger. Similarly, increasing the dimension of featurized representation $\phi_m$ also speeds up the convergence exponentially. They provide the guarantee for predictability even if $Q^*$ is not used. The proof is provided in Appendix E.

**Takeaway**. Featurzing pretrained representation as $\phi_m(h, Q)$ offers a simple and practical solution to balance *stability*, *predictability*, and *computability*. We just showed that $Q$ can simply be standard Gaussian distribution, and the dimension of $\phi_m(h)$ can be obtained by satisfying a specific approximation threshold $\epsilon$. It can also be treated as a tuning parameter in downstream tasks.

## 6 Benchmark and Real-world Experiments

We conduct experiments on the benchmark dataset *MovieLens-1m* (**ML-1m**) for *illustration* and *reproducibility* purposes. The *real-world production experiments* took place at a major US e-commerce platform anonymized as **Ecom**. The detailed descriptions for **ML-1m** and the introduction of **Ecom**'s production environment are provided in Appendix F.

**On ML-1m**. The dataset supports two types of pretraining-downstream task combination:
**(a).** leverage the sequences of user viewing data to pretrain movie embeddings, then use the embeddings to predict the genre of the movie (*ML-1m task 1*);
**(b).** pretrain movie embeddings using the title and other descriptions, then use the embeddings for downtream sequential recommendation (*ML-1m task 2*).

The detailed data processing, model and pretraining configurations, downstream training/testing setup, evaluation metrics, and **sensitivity analysis** are deferred to Appendix F. On *ML-1m task 1*, we use contrastive representation learning to pretrain the movie embeddings, and employ logistic regression to predict the genre using movie embeddings as features. On *ML-1m task 2*, we use a bidirectional-RNN-type structure on movies' NLP data, and extract the final hidden layer as pretrained representation. The downstream sequential recommendation task employs a *two-tower structure*, and a RNN is used to aggregate the history viewing sequence. In Table 1, we first see that $\phi_m(h)$ improves the stability of $h$ by at least $\times 10$ in both tasks. Even under the same dimension,

$\phi_m(h)$ outperforms $h$, and is highly comparable to $\text{avg}(h)$ – the manually stabilized version of $h$ by averaging it over ten independent runs. Note that $\text{avg}(h)$ is almost never a good practical solution because it requires repeating the same pretraining process multiple times. Here, we use it as an analytical baseline, and show that $\phi_m(h)$ is just as good. When the dimension increases for $\phi_m(h)$, it delivers much more superior results. Although changing dimension can also change the downstream model complexity, but as we discuss below, it offers more flexibility for real-world problems.

| | ML-1m Task1 | | ML-1m Task2 | | Ecom A/B testing | | | |
|---|---|---|---|---|---|---|---|---|
| | Accuracy | F1-Macro | Recall@5 | NDCG@5 | Task1 | Task2 | Task3 | Task4 |
| $h_{32}(x)$ | 64.82(2.83) | 44.79(2.21) | 66.55(3.03) | 33.96(1.83) | used as control | | | |
| $\text{avg}(h_{32}(x))$ | 68.72(-) | 59.04(-) | 68.98(-) | 35.02(-) | (not applicable) | | | |
| $\phi_{32}(\mathbf{h}(\mathbf{x}))$ | 68.64(0.27) | 58.91(0.25) | 68.90(.33) | 34.97(.17) | +0.24% | +0.13% | +0.14% | +0.10% |
| $\phi_{64}(\mathbf{h}(\mathbf{x}))$ | **85.77(0.19)** | **90.46(0.21)** | **71.30(.27)** | **37.48(.15)** | $p < 0.01$ | $p < 0.1$ | $p < 0.05$ | $p < 0.05$ |

Table 1: For **ML-1m** experiments, the results are multiplied by 100, and in the parenthesis are the standard deviations computed from ten independent runs. The subscripts of $h$ and $\phi$ refer to the dimension of representation, $\text{avg}(h)$ refers to using the representation averaged from the ten independent pretrainings. Note that $\text{avg}(h)$ cannot be used for large-scale production since it takes ten times the resources needed. For **Ecom A/B testing**, we present the relative lift over the control, and the p-value associated with the lift.

**On Ecom**. The item representation learning pipeline is being used by several downstream productions: *item-page recommendation* (*Task1*), *search ranking* (*Task2*), *email recommendation* (*Task3*), and *home-page marketing* (*Task4*). They all have task-specific features and non-trivial model architectures different. The refreshing of pretrained item embedding is done on a daily basis, and downstream model owners may have separate schedules to update and refresh the relevant parts of their models. In Appendix F.4, we describe our engineering solutions of deploying the featurization process on the frontend and backend.

During A/B testing, we observe performance lifts (in terms of *click-through rate*) that are statistically significant for *all four* downstream applications. The average *revenue-per-visitor* lift is also positive during the testing period. The detailed online results and analysis are provided in Appendix F.

**Lessons learnt**. In addition to improved stability and performance, an important feedback we received from downstream model owners is that the flexibility in choosing $\phi_m(h)$'s dimension is very helpful for their tasks. Prior to our featurization technique, it is almost impossible to personalize the dimension of pretrained representation for different applications, let alone tuning it in downstream tasks. Now knowing that the predictability will not vary much, experimenting with different dimensions often *allows them to find a better bias-variance tradeoff* for downstream tasks.

## 7 DISCUSSION

The analytical results and the proposed featurization method in our work can apply to a broad range of applications and research problems. Nevertheless, our results may still be rudimentary and far from providing the complete picture or optimal practice for using pretrained representation. We hope the progress we made will lead to more advanced future research and applications.

**Scope and limitation**. Most of our analysis are performed in basic settings: while they ensure the results will hold in generality, advanced methods for pretraining representation are not considered. Also, we do not include additional downstream features and their correlation with pretrained representations, or connections between the pretraining objective and downstream task. Those additional knowledge can be useful for deriving task-specific results (Arora et al., 2019). For application, our featurization technique may be less helpful if the downstream task simply uses embedding distance like *KNN search*. Optimizing the space and time complexity by such as *embedding quantization* might be more useful for such tasks (Chen et al., 2020), which is not discussed in our paper.

**A future direction**. While our work studies $h(x)$ as a whole, it can be inferred from Figure 1c that the element-wise variance of $\hat{h}(x)$ is bimodal, which suggests heterogeneity within $h(x)$. Possible explanations are that a (random) subset of $h(x)$ is responsible for *overfitting* the pretraining task (Bartlett et al., 2020), or that some dimensions are forced to become more independent of others so the representation matrix has nice *spectral properties* (Hastie et al., 2022). It is thus an important future direction to identify the internal structure of $h(x)$ to better featurize pretrained representations.

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

## A    TECHNICAL BACKGROUND

In this part of the paper we provide the technical background for both the discussions in the paper and the following proofs.

A central role for proving uniform convergence results is the Gaussian / Rademarcher complexity. For a set $A \subset \mathbb{R}^n$, it is defined as:

$$\mathcal{G}(A) := \mathbb{E}_\epsilon \Big[ \sup_{a \in A} \sum_{i=1}^{n} a_i \Big],$$

where $\epsilon$ are $i.i.d$ Gaussian / Rademarcher random variables. It essentially measures how good a function class can interpolate a random sign pattern assigned to a set of points. Given a function class $\mathcal{F}$ and $n$ samples $(x_1, \ldots, x_n)$, the empirical Gaussian / Rademarcher complexity is given by:

$$\mathcal{G}_n(\mathcal{F}) := \mathbb{E}_\epsilon \Big[ \sup_{f \in \mathcal{F}} \sum_{i=1}^{n} f(x_i) \Big].$$

**Remark A.1.** *We mention that in some versions of the definition, there is a $1/n$ factor in the complexity term. Here, we explicit pull that factor out and place it in the resulting bound.*

As we mentioned earlier, an important reason for us using Gaussian complexity is because some of its technical properties, which is the *Slepian's Lemma* (Slepian, 1962) and its corollary, which we state as below:

**Lemma A.1** (From Slepian's Lemma). *Suppose $\phi : A \to \mathbb{R}^q$ has Lipschitz constant L. Then it holds that:*

$$\mathcal{G}(\phi(A)) \leq L\mathcal{G}(A).$$

This result can be viewed as the contraction lemma for Gaussian complexity (Ledoux & Talagrand, 1991).

### A.1    INDUCTIVE BIAS OF GRADIENT DESCENT

Our introduction primarily follows Soudry et al. (2018); Ji & Telgarsky (2019); **?**); Gunasekar et al. (2018) and their follow-up works. The key factor that contributes to the implicit bias of gradient decent is the divergence of model parameters after separating the data under loss functions that has exponential-tail behavior.

When the predictor $f \in \mathcal{F}$ parameterized by $\theta$ is over-parameterized, other than certain degenerated cases, the data can be separated at certain point if the predictor class satisfies some regularity assumptions (Lyu & Li, 2019), e.g.

- $f \in \mathcal{F}$ is homogeneous such that: $f(x; c \cdot \theta) = c^\beta f(x; \theta), \forall c > 0$;
- $f \in \mathcal{F}$ is smooth and has bounded Lipschitz constant.

These conditions can be met for many neural network structures and activation functions. The exponential-tail of loss function can be satisfied by the common exponential loss and logistic loss (which we use through our discussions and experiments).

To see why the the norm of model parameters will diverge, simply note that under such as exponential loss, both the risk and the gradient will take the form of: $\sum_i c_i \exp(-y_i f(x_i; \theta))$, where $c_i$ are lower order terms. Since gradient descent will converge to a stationary point due to the nice

properties of $\mathcal{F}$, we expect $\sum_i c_i \exp(-y_i f(x_i; \theta)) = 0$ to hold upon convergence. A necessary condition for that is: $\exp(-y_i f(x_i; \theta)) = 0, i = 1, \ldots, n$, and this condition is actually sufficient with high probability (Soudry et al., 2018).

Therefore, for all $\exp(-y_i f(x_i; \theta))$ to reach 0, $\|\theta\|$ must diverge so $|f(\cdot; \theta)| \to \infty$. With that said, since the loss function decays exponentially fast around 0, **the data points with the largest *margin* will dominate both the gradient and the loss function**.

As a direct consequence, the decision boundary will share characteristics with the hard-margin SVM, given by:

$$\min \|\theta\|_2 \quad s.t. \quad y_i f(x_i; \theta) \geq 1, \ \forall i = 1, \ldots, n.$$

Indeed, recent work shows that the optimization path of over-parameterized models will indeed converge to some minimum-norm predictor:

**Corollary A.1** (Chizat et al. (2019); Woodworth et al. (2020), and others). *Under the conditions specified in the reference work, which are mostly exponential loss, scaled initialization, appropriate learning rate, and regularity conditions for the predictor class, it holds that:*

$$\lim_{t \to \infty} \lim_{d \to \infty} F\big(\theta^{(t)}/\|\theta^{(t)}\|\big) \overset{\text{stationary points of}}{\to} \Big\{ \arg\min \|f\|_K \ s.t. \ y_i f(x_i) \geq 1, \ \forall i \in [n] \Big\},$$

*where $F$ is the decision boundary of $f$, $d$ is the dimension of the hidden layer(s) of $f$, and $\|\cdot\|_K$ is an appropriate RKHS norm.*

Note that in Section 4.2 we use $g_0$ to denote the converged result, and the above corollary guarantees its existence and uniqueness. However, one open question is which particular RKHS norm best describes the solution, because it will particularly affect the convergence of the parameters. Therefore, in our work, we leave the convergence of parameters out of our discussion.

**Remark A.2.** *It is also worth mentioning that the convergence of $\mathbb{E}[h^{(t)}(x)]$ plays no part in our arguments and results. Indeed, it will not change the stochasticity of $h^{(t)}(x)$, and (in some cases) can be implied from the convergence of $g^{(t)}(x)$ (Lyu & Li, 2019). Therefore, we do not discuss it in our work.*

## B  PROOF OF THE RESULTS IN SECTION 4.1

We prove Proposition 1 in this part of the appendix. An important result we will be using is the Gaussian complexity bound for empirical risk minimization, and we will use the version of Bartlett & Mendelson (2002).

**Lemma A.2.** *Let $\mathcal{F}$ be real-valued function class from $\mathcal{X}$ to $[0, 1]$. Let $(X_1, \ldots, X_n)$ be i.i.d random variables, then for all $f \in \mathcal{F}$, it holds with probability at least $1 - \delta$ that:*

$$\mathbb{E}\big[f(X)\big] \leq \frac{1}{n} \sum_i f(X_i) + \frac{\sqrt{2\pi}\mathcal{G}_n(\mathcal{F})}{n} + \sqrt{\frac{9 \log 2/\delta}{2n}}.$$

We now provide the proof, part of which will be using Corollary Lemma A.1, and Lemma A.2. We also assume $\mathcal{F}$ has a Lipschitz constant of at most $L$.

*Proof.* Recall that $h^*, f^* := \arg\min_{h \in \mathcal{H}, f \in \mathcal{F}} R(h, f)$. We decompose the generalization error via:

$$
\begin{aligned}
R(\hat{h}) - \min_{h \in \mathcal{H}, f \in \mathcal{F}} R(h, f) = &\left( R(\hat{h}) - \min_{f \in \mathcal{F}} \frac{1}{n} \sum_i \ell\big(f \circ \hat{h}(X_i), Y_i\big) \right) + \\
&+ \left( \min_{f \in \mathcal{F}} \frac{1}{n} \sum_i \ell\big(f \circ \hat{h}(X_i), Y_i\big) - \min_{f \in \mathcal{F}} \frac{1}{n} \sum_i \ell\big(f \circ h^*(X_i), Y_i\big) \right) \\
&+ \left( \min_{f \in \mathcal{F}} \frac{1}{n} \sum_i \ell\big(f \circ h^*(X_i), Y_i\big) - \mathbb{E}_{P^n}\Big[ \min_{f \in \mathcal{F}} \frac{1}{n} \sum_i \ell\big(f \circ h^*(X_i), Y_i\big) \Big] \right) \\
&+ \left( \mathbb{E}_{P^n}\Big[ \min_{f \in \mathcal{F}} \frac{1}{n} \sum_i \ell\big(f \circ h^*(X_i), Y_i\big) \Big] - \min_{f \in \mathcal{F}} \mathbb{E}_{(X,Y) \sim P} \ell\big(f \circ h^*(X), Y\big) \right).
\end{aligned}
\tag{A.1}
$$

We first discuss the first term, which incurs a major discussion in Section 4.1. By a standard practice, the first term can be bounded via:

$$R(\hat{h}) - \min_{f \in \mathcal{F}} \frac{1}{n} \sum_i \ell\big(f \circ \hat{h}(X_i), Y_i\big)$$

$$\leq \sup_{h \in \mathcal{H}} \left\{ R(\hat{h}) - \min_{f \in \mathcal{F}} \frac{1}{n} \sum_i \ell\big(f \circ \hat{h}(X_i), Y_i\big) \right\}$$

$$\leq \sup_{h \in \mathcal{H}} \mathbb{E}_{P^n} \Big[ \mathbb{E}_{(X,Y)\sim P} \big[ \ell\big(f_{h,n} \circ h(X), Y\big) - R_n(h) \big] \Big] \quad \text{(a)}$$

$$\sup_{h \in \mathcal{H}} \left\{ \mathbb{E}_{P^n} \Big[ \frac{1}{n} \sum_i \ell\big(f_{h,n} \circ h(X_i), Y_i\big) \Big] - \frac{1}{n} \sum_i \ell\big(f_{h,n} \circ h(X_i), Y_i\big) \right\} \quad \text{(b)}$$

Using Lemma A.2, term (b) can be bounded as:

$$\sup_{h \in \mathcal{H}} \left\{ \mathbb{E}_{P^n} \Big[ \frac{1}{n} \sum_i \ell\big(f_{h,n} \circ h(X_i), Y_i\big) \Big] - \frac{1}{n} \sum_i \ell\big(f_{h,n} \circ h(X_i), Y_i\big) \right\} \leq \sqrt{2\pi} \mathcal{G}_n(\mathcal{A}(\mathcal{H})) + \sqrt{9 \log 2/\delta},$$

where the set $\mathcal{A}(\mathcal{H})$ is given by:

$$\left\{ \left( \frac{1}{n} \ell\big(f_{h,n} \circ h(X_1), Y_1\big), \ldots, \frac{1}{n} \ell\big(f_{h,n} \circ h(X_n), Y_n\big) \right) : h \in \mathcal{H} \right\}.$$

It is easy to examine that $\mathcal{A}(\mathcal{H})$ invokes Slepian's lemma, so we can use the contraction result from Lemma A.1 to further bound it:

$$\mathcal{G}_n(\mathcal{A}(\mathcal{H})) \leq \frac{L}{\sqrt{n}} \mathcal{G}_n(\mathcal{H}).$$

Combined together, the term (b) is upper bounded as: $\frac{\sqrt{2\pi} L \mathcal{G}_n(\mathcal{H})}{\sqrt{n}} + \sqrt{9 \log 2/\delta}$.

Now we bound term (a) as below. Define the shorthand $\ell(\mathcal{F}(h)) := \big\{ \ell\big(f(h(X_1), Y_1), \ldots, \ell(f(h(X_n), Y_n))\big) : f \in \mathcal{F} \big\}$. It holds that:

$$\sup_{h \in \mathcal{H}} \mathbb{E}_{P^n} \Big[ \mathbb{E}_{(X,Y)\sim P} \big[ \ell\big(f_{h,n} \circ h(X), Y\big) - R_n(h) \big] \Big]$$

$$\leq \sup_{h \in \mathcal{H}} \mathbb{E}_{P^n} \sup_{f \in \mathcal{F}} \left\{ \mathbb{E}_{(X,Y)\sim P} \ell\big(f \circ h(X), Y\big) - \frac{1}{n} \sum_i \ell\big(f \circ h(X_i), Y_i\big) \right\}$$

$$\leq \sqrt{2\pi} \sup_{h \in \mathcal{H}} \mathbb{E}_{P^n} \frac{\mathcal{G}_n(\ell(\mathcal{F}(h)))}{n} \quad \text{(using Lemma A.2) and A.1} \tag{A.2}$$

$$= \sqrt{2\pi} n^{-1} \sup_{h \in \mathcal{H}} \mathbb{E}_{P^n} \frac{\mathcal{G}_n(\ell(\mathcal{F}(h)))}{\|h(X)\|} \|h(X)\| \quad \text{(where } h(X) := [h(X_1), \ldots, h(x_n)])$$

$$\leq \sqrt{2\pi} n^{-1} \sup_{h \in \mathcal{H}} \sqrt{\mathbb{E}\|h(X)\|^2} \cdot \sup_{A \in \mathbb{R}^{n \times d}} \frac{1}{\|A\|} \mathbb{E} \sup_{f \in \mathcal{F}} \sum_i \epsilon_i f([A]_i) \quad \epsilon_i \overset{i.i.d}{\sim} N(0,1).$$

We let $\mathcal{G}'_n(\mathcal{F}) := \sup_{A \in \mathbb{R}^{n \times d}} \frac{1}{\|A\|} \mathbb{E} \sup_{f \in \mathcal{F}} \sum_i \epsilon_i f([A]_i)$ be the modified Gaussian complexity, so the term (b) is finally bounded by: $\frac{\sqrt{2\pi}}{n} \mathcal{G}'_n(\mathcal{F}) \sup_{h \in \mathcal{H}} \sqrt{\mathbb{E}\|h(X)\|^2}$.

Next, notice in the last term that:

$$\mathbb{E}_{P^n} \Big[ \min_{f \in \mathcal{F}} \frac{1}{n} \sum_i \ell\big(f \circ h^*(X_i), Y_i\big) \Big] \leq \mathbb{E}_{P^n} \frac{1}{n} \sum_i \ell\big(f^* \circ h^*(X_i), Y_i\big)$$

$$= \mathbb{E}_{(X,Y)\sim P} \ell\big(f^* \circ h^*(X), Y\big).$$

Therefore, the last term is always non-positive.

Similar, by definition, the second term is non-positive as well.

Finally, as for the second term, since there is already non-concentrating terms appearing in the bound of the first term, it does not hurt to simply bound it using Hoeffding's bound, i.e. the first term will not exceed $\mathcal{O}(\sqrt{\log 1/\delta})$ with probability at least $1 - \delta$. Putting things together, we conclude the final result.

$\square$

## C  TECHNICAL DETAILS FOR SECTION 4.2

We first restate the proposition:

**Proposition A.1.** *For the one-layer MLP we consider, with the learning rate $\alpha = 1$, for any step $t > 1$, as $d \to \infty$, the model output $g^{(t)}(x)$ will converge almost surely to $g_*^{(t)}(x)$ defined as follows:*

$$g_*^{(t)}(x) = (C_1^{(t)}C_2^{(t)} + C_3^{(t)}C_4^{(t)})x,$$

*with* $(C_1^{(t+1)}, C_2^{(t+1)}, C_3^{(t+1)}, C_4^{(t+1)}) = (C_1^{(t)}, C_2^{(t)}, C_3^{(t)}, C_4^{(t)}) + \mathcal{L}^{(t)}x^{(t)}(C_3^{(t)}, C_4^{(t)}, C_1^{(t)}, C_2^{(t)}).$

The above iterative update result can be shown by making explicit of the terms following the forward and backward pass in $t^{th}$ gradient step.

In particular, it holds that:

$$g^{(t)}(x) \overset{a.s.}{\to} \mathbb{E}\Theta^{(t)}\sigma(W^{(t)}x) \quad (\overset{\text{def}}{=} g_*^{(t)}(x)),$$
$$\ell'(g^{(t)}(x^{(t)}), y^{(t)}) \overset{a.s.}{\to} \ell'(g_*^{(t)}(x^{(t)}), y^{(t)})\mathcal{L}^{(t)} \quad (\overset{\text{def}}{=} \mathcal{L}^{(t)}),$$
$$\tilde{\Theta}^{(t+1)} = \tilde{\Theta}^{(t)} - \mathcal{L}^{(t)}\sigma(\tilde{W}^{(t)}x^{(t)}),$$
$$\tilde{W}^{(t+1)} = \tilde{W}^{(t)} - \mathcal{L}^{(t)}x^{(t)}\tilde{\Theta}^{(t)}\sigma'(\tilde{W}^{(t)}x^{(t)}).$$

The only extra requirement for the above convergence to hold is that the activation function is well-behaved (see Yang (2019) for a detailed description).

To see how the above system of equations lead to the result in Proposition A.1, imagine the activation is the identity function. In this case, $\tilde{\Theta}^{(t)}$ and $\tilde{W}^{(t)}$ are always deterministic linear combinations of $\tilde{\Theta}^{(0)}$ and $\tilde{W}^{(0)}$. Observe that the update becomes:

$$\tilde{\Theta}^{(t)} = C_1\tilde{\Theta}^{(0)} + C_2\tilde{W}^{(0)}, \quad \tilde{W}^{(t)} = C_3\tilde{\Theta}^{(0)} + C_4\tilde{W}^{(0)}.$$

We mention that as a corollary, $\mathbf{W}^{(t+1)}(x)$ is also element-wise $i.i.d$, so the inner product of the hidden representations

$$\langle \mathbf{W}^{(t+1)}(x), \mathbf{W}^{(t+1)}(x') \rangle \overset{a.s.}{\to} \mathbb{E}[W^{(t)}x \cdot W^{(t)}x'],$$

where $W^{(t)}$ is an $i.i.d$ row of $\tilde{W}^{(t+1)}$, which is the rescaled version of $\mathbf{W}^{(t+1)}$.

## D  PROOFS OF THE RESULTS IN SECTION 4.3

**Proof for Proposition 3**

*Proof.* The proofs for the lower bound often starts by converting the problem to a hypothesis testing task. Denote our parameter space by $\mathcal{B}(k) = \{\theta \in \mathbb{R}^d : \|\theta\|_0 \le k\}$.

The intuition is that suppose the data is generated by:

1. drawing $\theta$ according to an uniform distribution on the parameter space;

2. conditioned on the particular $\theta$, the observed data is drawn.

Then the problem is converted to determining according to the data if we can recover the underlying $\theta$ as a canonical hypothesis testing problem.

For any $\delta$-packing $\{\theta_1, \ldots, \theta_M\}$ of $\mathcal{B}(k)$, suppose $B$ is sampled uniformly from the $\delta$-packing. Then following a standard argument of the *Fano method* Wainwright (2019), it holds that:

$$P\big(\min_{\hat{\theta}} \sup_{\|\theta^*\|_0 \le k} \|\hat{\theta} - \theta^*\|_2 \ge \delta/2\big) \ge \min_{\tilde{\theta}} P\big(\tilde{\theta} \ne B\big), \tag{A.3}$$

where $\tilde{\theta}$ is a testing function that decides according to the data if the some estimated $\theta$ equals to an element sampled from the $\delta$-packing.

The next step is to bound $\min_{\tilde{\theta}} P\big(\tilde{\theta} \ne B\big)$, whereas by the information-theoretical lower bound (Fano's Lemma), we have:

$$\min_{\tilde{\theta}} P\big(\tilde{\theta} \ne B\big) \ge 1 - \frac{I(y, B) + \log 2}{\log M}, \tag{A.4}$$

where $I(\cdot, \cdot)$ denotes the mutual information.

Then we only need to bound the mutual information term. Let $P_\theta$ be the distribution of $\mathbf{y}$ (which the vector consisting of the $n$ samples) given $B = \theta$. Since $\mathbf{y}$ is distributed according to the mixture of: $\frac{1}{M} \sum_i P_{\theta_i}$, it holds:

$$I(y, B) = \frac{1}{M} \sum_i D_{KL}\big(P_{\theta_i} \| \frac{1}{M} \sum_j P_{\theta_j}\big) \le \frac{1}{M^2} \sum_{i,j} D_{KL}\big(P_{\theta_i} \| P_{\theta_j}\big),$$

where $D_{KL}$ is the Kullback-Leibler divergence. The next step is to determine $M$: the size of the $\delta$-packing, and the upper bound on $D_{KL}\big(P_{\theta_i} \| P_{\theta_j}\big)$ where $P_{\theta_i}, P_{\theta_j}$ are elements of the $\delta$-packing.

For the first part, it has been shown that there exists a $1/2$-packing of $\mathcal{B}(k)$ in $\ell_2$-norm with $\log M \ge \frac{k}{2} \log \frac{d-k}{k/2}$ (Raskutti et al., 2011).

As for the bound on the KL-divergence term, note that given $\theta$, $P_\theta$ is a product distribution of the condition Gaussian: $y|\epsilon \sim N\big(\theta^\intercal \epsilon \frac{\sigma_h^2}{\sigma_\epsilon^2}, \theta^\intercal \theta(\sigma_z^2 - \sigma_z^4/\sigma_\epsilon^2)\big)$, where $\sigma_\epsilon^2 := \sigma_h^2 + \sigma_\epsilon^2$.

Henceforth, for any $\theta_1, \theta_2 \in \mathcal{B}(k)$, it is easy to compute that:

$$D_{KL}(P_{\theta_1} \| P_{\theta_2})$$
$$= \mathbb{E}_{P_{\theta_1}} \left[ \frac{n}{2} \log \left( \frac{\theta_1^\intercal \theta_1 (\sigma_z^2 - \sigma_z^4/\sigma_\epsilon^2)}{\theta_2^\intercal \theta_2 (\sigma_z^2 - \sigma_z^4/\sigma_\epsilon^2)} \right) + \frac{\big\| \mathbf{y} - \theta_2^\intercal \boldsymbol{\epsilon} \frac{\sigma_h^2}{\sigma_\epsilon^2} \big\|_2^2}{2\theta_2^\intercal \theta_2 (\sigma_z^2 - \sigma_z^4/\sigma_\epsilon^2)} - \frac{\big\| \mathbf{y} - \theta_1^\intercal \boldsymbol{\epsilon} \frac{\sigma_h^2}{\sigma_\epsilon^2} \big\|_2^2}{2\theta_1^\intercal \theta_1 (\sigma_z^2 - \sigma_z^4/\sigma_\epsilon^2)} \right]$$
$$= \frac{\sigma_z^2}{2\sigma_\epsilon^2} \|\boldsymbol{\epsilon}(\theta_1 - \theta_2)\|_2^2,$$

where $\mathbf{y}$ and $\boldsymbol{\epsilon}$ are the vector and matrix consists of the $n$ samples, i.e. $\mathbf{y} \in \mathbb{R}^n$ and $\boldsymbol{\epsilon} \in \mathbb{R}^{n \times d}$.

Since each row in the matrix $\boldsymbol{\epsilon}$ is drawn from $N(0, \sigma_\epsilon^2 I_{d \times d})$, standard concentration result shows that with high probability, $\|\boldsymbol{\epsilon}(\theta_1 - \theta_2)\|_2^2$ can be bounded by $C\|\theta_1 - \theta_2\|_2^2$ for some constant $C$. It gives us the final upper bound on the KL divergence term:

$$D_{KL}(P_{\theta_1} \| P_{\theta_2}) \lesssim \frac{n\sigma_z^2 \delta^2}{2\sigma_\epsilon^2}.$$

Substituting this result into (A.4) and (A.3), by choosing $\delta^2 = \frac{Ck\sigma_\epsilon^2}{\sigma_z^2 n} \log \frac{d-k}{k/2}$ and rearranging terms, we obtain the desired result that with probability at least $1/2$:

$$\inf_{\hat{\theta}} \sup_{\theta^*: \|\theta^*\|_0 \le k} \|\hat{\theta} - \theta^*\|_2 \gtrsim \frac{\sigma_\epsilon^2}{\sigma_h^2} kn^{-1} \log(d/k).$$

$\square$

**Proof of Lemma 1**

*Proof.* We first express the NW predictor in its expectation form:

$$f_\phi(X) = \frac{E_{X'}\big[y'K(X,X')\big]}{Z},$$

where $Z$ is the normalization constant. Recall that $y \in \{-1,+1\}$, $R(\cdot)$ is risk associated with the $0-1$ classification loss. We first define for $x \in \mathcal{X}$:

$$\gamma_\phi(X) := \sqrt{\frac{\mathbb{E}_{X'}\big[K(X,X')\big]}{Z}},$$

where the expectation is taken w.r.t. the underlying distribution. Using the Markov inequality, we immediately have: $|\gamma(X)| \le \frac{1}{\sqrt{\delta}}$ with probability at least $1-\delta$. It then holds that:

$$
\begin{aligned}
1 - R(f) &= P\big(Yf(X) \ge 0\big)\\
&\ge \mathbb{E}\Big[\frac{Yf(X)}{\gamma(X)} \cdot \mathbb{1}[Yf(X) \ge 0]\Big]\\
&\ge \mathbb{E}\Big[\frac{Yf(X)}{\gamma(X)}\Big] \ge \frac{\mathbb{E}\big[\mathbb{1}[Y=Y']K(X,X')\big]}{Z}\sqrt{\delta} \text{ ,with probability } 1-\delta,
\end{aligned}
$$

which concludes the proof. $\qquad\square$

## E    PROOF OF THE RESULT IN SECTION 5

The proof of Proposition 4 relies on two important results, which we state below.

**Lemma A.3** (Ben-Tal et al. (2013)). *Let $c$ be any closed convex function with domain $[0,+\infty)$, and this conjugate is given by $c^*(s) = \sup_{t\ge 0}\{ts - c(t)\}$. Then for any distribution $Q^*$ and any function $g : \mathbb{R}^d \to \mathbb{R}$, it holds:*

$$\sup_{Q \in \mathcal{Q}(Q^*;\delta)} \int g(u)dQ(u) = \inf_{\lambda \ge 0,\eta} \Big\{\lambda \int c^*\Big(\frac{g(u)-\eta}{\lambda}\Big)dQ^*(u) + \delta\lambda + \eta\Big\}. \tag{A.5}$$

The next lemma is adapted from the concentration of random Fourier feature in Rahimi & Recht (2007). Recall that $\phi_m\big(h(x),Q\big) := 1/\sqrt{m}\big[\cos\big(u_1 h(x)\big), \sin\big(u_1 h(x)\big), \ldots, \cos\big(u_m h(x)\big), \sin\big(u_m h(x)\big)\big]$ comes from the Monte Carlo approximation of $K(h(x),h(x'))$.

**Lemma A.4.** *Let $\mathcal{A} \subset \mathbb{R}^d$ has diameter $d_A$ such that $h(x) \in A$ for all $x \in \mathcal{X}$. It holds that:*

$$
\begin{aligned}
Pr\Big(&\sup_{h(x),h(x')} \big|K(h(x),h(x')) - \langle\phi_m\big(h(x),Q\big), \phi_m\big(h(x'),Q\big)\rangle\big| \ge \epsilon\Big)\\
&\le 2^8\Big(\frac{\sigma_Q d_A}{\epsilon}\Big)\exp\Big(-\frac{m\epsilon^2}{4(d+2)}\Big),
\end{aligned}
\tag{A.6}
$$

*where $Q$ is given by the inverse Fourier transform of $K$, and $\sigma_Q$ is the second moment of $Q$.*

Recall that $A_n(Q) := \frac{1}{n(n-1)}\sum_{i\ne j}\mathbb{1}[y_i = y_j]\langle\phi_m(h(x_i),Q),\phi_m(h(x_i),Q)\rangle$. For notation convenience, in what follows, we let $h_i := h(x_i)$, and further define $\tilde{\phi}(h,U) := [\cos(U^T h),\sin(U^T h)]$ as the actual random Fourier feature underlying $\phi_m(h,Q)$, where $U \sim Q$. Also, we let $K(Y,Y') := 1[Y=Y']$ to be the labelling kernel of the downstream task.

*Proof.* Following Lemma A.3, we work with a scaled version of the f-divergence under $c(t) = \frac{1}{k}(t^k - 1)$ (because its dual function has a cleaner form). It is easy to check that $c^*(s) = \frac{1}{k'}[s]_+^{k'} + \frac{1}{k}$ with $\frac{1}{k'} + \frac{1}{k} = 1$.

First note that the sampling error of the alignment $\mathbb{E}\big[K(Y_i,Y_j)K_Q(H_i,H_j)\big]$, i.e. replacing the expectation by the sample average, can be given by:

$$\Delta_n(U) := \frac{1}{n(n-1)} \sum_{i \neq j} K(y_i, y_j) \tilde{\phi}(h_i, U)^T \tilde{\phi}(h_j, U) - \mathbb{E}\big[K(Y_i, Y_j) K_Q(H_i, H_j)\big]$$

$$= \frac{1}{n(n-1)} \sum_{i \neq j} K(y_i, y_j) \tilde{\phi}(h_i, U)^T \tilde{\phi}(h_j, U) - \mathbb{E}\big[K(Y_i, Y_j) \tilde{\phi}(H_i, U)^T \tilde{\phi}(H_i, U)\big].$$

We show that $\Delta_n(U)$ is sub-Gaussian. Let $\{h_i', y_i'\}_{i=1}^n$ be an $i.i.d$ copy off the observation expect for one element such that $(h_j, y_j) \neq (h_j', y_j')$. Without loss of generality, we assume the last element is different: $(h_n, y_n) \neq (h_n', y_n')$. Let $\Delta_n'(U)$ be computed by replace $\{h_i, y_i\}_{i=1}^n$ with $\{h_i', y_i'\}_{i=1}^n$, and their difference can be bounded via:

$$|\Delta_n(U) - \Delta_n'(U)|$$

$$= \frac{1}{n(n-1)} \Big| \sum_{i \neq j} K(y_i, y_j) \tilde{\phi}(h_i, U)^T \tilde{\phi}(h_j, U) - K(y_i', y_j') \tilde{\phi}(h_i', U)^T \tilde{\phi}(h_j', U) \Big|$$

$$\leq \frac{1}{n(n-1)} \Big( \sum_{i < n} \big| K(y_i, y_n) \tilde{\phi}(h_i, U)^T \tilde{\phi}(h_n, U) - K(y_i, y_n') \tilde{\phi}(h_i, U)^T \tilde{\phi}(h_n', U) \big|$$

$$+ \sum_{j < n} \big| K(y_n, y_j) \tilde{\phi}(h_n, U)^T \tilde{\phi}(h_j, U) - K(y_n', y_j) \tilde{\phi}(h_n', U)^T \tilde{\phi}(h_j, U) \big| \Big)$$

$$\leq \frac{4}{n}$$

where the last inequality comes from the fact that the random Fourier features $\tilde{\phi}$ and the labelling kernel $K(y, y')$ are both bounded by 1. Therefore, the above bounded difference result tells that $\Delta_n(U)$ is a $\frac{4}{n}$-subGaussian random variable.

To bound $\Delta_n(U)$, we use:

$$\sup_{Q \in \mathcal{Q}(Q^*; \delta)} \Big| \int \Delta_n(U) dQ \Big| \leq \sup_{Q \in \mathcal{Q}(Q^*; \delta)} \int |\Delta_n(U)| dQ$$

$$\leq \inf_{\lambda \geq 0} \Big\{ \frac{\lambda^{1-k'}}{k'} \mathbb{E}_{Q^*}\big[|\Delta_n(U)|^{k'}\big] + \frac{\lambda(\delta+1)}{k} \Big\} \text{ (using Lemma A.3)}$$

$$= (\delta+1)^{1/k} \mathbb{E}_{Q^*}\big[|\Delta_n(U)|^{k'}\big]^{1/k'} \text{ (solving for } \lambda^* \text{ from above)}$$

$$= \sqrt{\delta+1} \mathbb{E}_{Q^*}\big[|\Delta_n(U)|^2\big]^{1/2} \quad \text{(let } k = k' = 1/2\text{).}$$

$$\tag{A.7}$$

It means that in order to bound $\sup_{Q \in \mathcal{Q}(Q^*; \delta)} \big| \int \Delta_n(U) dQ \big|$, we only need to bound $|\Delta_n(U)|^2$. Using classical results for sub-Gaussian random variables (Boucheron et al., 2013), it holds for $\lambda \leq n/8$ that:

$$\mathbb{E}\big[ \exp\big(\lambda \Delta_n(U)\big)^2 \big] \leq \exp\big(-\frac{1}{2} \log(1 - 8\lambda/n)\big).$$

We can take its integral and further upper bound the result with:

$$p\Big( \int \Delta_n(U)^2 dQ \geq \frac{\epsilon^2}{\delta+1} \Big)$$

$$\leq \mathbb{E}\Big[ \exp\Big(\lambda \int \Delta_n(U)^2 dQ\Big)\Big] \exp\Big(-\frac{\lambda \epsilon^2}{\delta+1}\Big) \quad \text{(Chernoff bound)}$$

$$\leq \exp\Big(-\frac{1}{2} \log\Big(1 - \frac{8\lambda}{n}\Big) - \frac{\lambda \epsilon^2}{\delta+1}\Big) \quad \text{(apply Jensen's inequality).}$$

Hence, it holds that:

$$Pr\Big( \sup_{Q \in \mathcal{Q}(Q^*; \delta)} \Delta_n(U) \geq \epsilon \Big) \leq \exp\Big(-\frac{n\epsilon^2}{16(1+\delta)}\Big).$$

Combine this result with the approximation error of random Feature feature in Lemma A.4, we obtain the desired result. $\qquad \square$

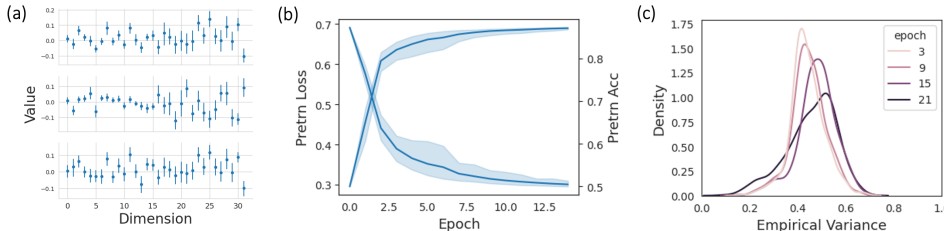

Figure A.1: Replicating the results shown in Figure 1a, 1b, and 1c on IMDB dataset. (a). The dimension-wise variance of the pretrained embedding values of three randomly sampled reviews; (b). The convergence of the pretraining loss and accuracy. (c). The distribution of the empirical variances of the pretrained embedding values, as pretraining progresses.

# F    SUPPLEMENTARY MATERIAL FOR THE EXPERIMENTS

We provide the descriptions, details, and additional results of our experiments in this part of the appendix.

## F.1    REPLICATING THE INSTABILITY ISSUE WITH IMDB DATASET

The **IMDB** dataset is a binary sentiment analysis dataset consisting of 50,000 reviews from the Internet Movie Database (IMDb) labeled as positive or negative[4]. We particularly consider using this dataset for an addition proof of concept because it appears on the official *tutorial of Tensorflow*[5].

We directly adopt the implementation from the tutorial, including the text prepossessing pipeline and model architecture. In particular, the raw input texts are pass to a text vectorization layer, an embedding layer, a bidirectional RNN layer, and finally two dense layers to produce the final score for binary classification. We extract the hidden representation from the last hidden layer as the hidden representation. In our experiments, we set the number of hidden dimension as 32.

The results are provided in Figure A.1, where we observe patterns highly similar to the ML-1m data. In particular, the pretrained embeddings both have high variances in their exact values even if their pretraining objectives converge to similar loss and accuracy, and the variances gets larger as the pretraining progress. Two minor differences from the ML-1m result are that the pretraining process is less stable for IMDB (Figure A.1b), and the variance distribution here is unimodal instead of the bimodal distribution we observed in Figure 1c.

## F.2    DETAILS OF THE BENCHMARK EXPERIMENTS

The main benchmark experiments in our paper are conducted on the **Movielens-1m**[6] dataset, which is a well-established public dataset for movie & user contextual analysis and examining recommendation.

The **ML-1m** dataset consists of 1 million movie ratings from 6,000 users on 4,000 movies, with a one-to-five rating scale. According to Harper & Konstan (2015), the data is collected from the initial and follow-up stages, where the initial stage mainly involves popularity-based exposure (a very small proportion involves random exposure), while in the follow-up stage, rating feedback is collected under some deterministic recommender systems. By convention, we convert the dataset to **implicit feedback**, which amounts to treating all rating events as clicks. For contextual information, each movie is provided with its title and genre, in the form of English words or sentences. There are 18 genres in total.

**Pretraining movie embeddings from user behavior data**

---

[4]https://www.imdb.com/interfaces/
[5]https://www.tensorflow.org/text/tutorials/text_classification_rnn
[6]https://grouplens.org/datasets/movielens/1m/

We use **Item2vec** (Barkan & Koenigstein, 2016) to train movie embedding from users' consecutive viewing data. Item2vec uses the same objective function as Word2vec (Mikolov et al., 2013)j, where the words become movies and the corpus become each user's viewing sequence. Movies belong to a consecutive viewing window of #ws are treated as positive pairs, and for each positive pair, we randomly sample #ns negative movies. Together with the embedding dimension $d$ and $\ell_2$-regularization parameter (weight decay) $\lambda$, the training schema is described by the quadruplet of (#ws, #ns, $d$, $\lambda$). Since our goal is not to find the best pretraining schema, we fix #ws=3 and #ns=3, and focus on studying how the our results may change under different $d$.

**Pretraining movie embeddings from movie contextual data**

Since the movie titles and other contextual information can be relatively short, large NLP models may not be appropriate. Therefore, we use the Doc2vec model Dai et al. (2015) to pretrain the movie embeddings. Since Doc2vec is built on top of Word2vec, the training schema can also be described by the quadruplet of (#ws, #ns, $d$, $\lambda$). Therefore, we also #ws=3 and #ns=3.

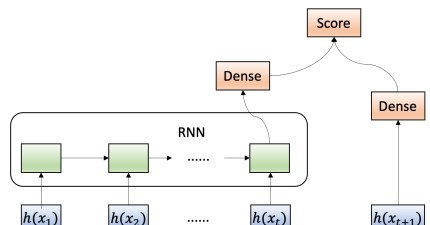

Figure A.2: The two-tower architecture.

**Using pretrained movie embedding for downstream genre prediction**

Given pretrained movie embeddings $\hat{h}(x)$, we employ logistic regression to predict the score for the movie to belong to a particular genre, i.e. $p(Y_i = k) \propto \exp(\theta_k \hat{h}(x))$. Due to its simplicity, we use the logistic regression subroutine from the *scikit-learn* package.

**Using pretrained movie embedding for downstream sequential recommendation**

We employ a two-tower model structure (Figure A.2) for the downstream sequential recommendation, which is very common in the recommendation community. In particular, we use RNN to aggregate the past interaction sequence, so the whole model is very similar to *GRU4Rec* (Jannach & Ludewig, 2017). We use the sigmoid function as the activation function for the dense layers. The model training can be done in a *seq-to-seq* fashion, where for each positive target, we randomly sample 3 negative targets. We fix the hidden dimension of both the RNN and the dense layers as 16.

**Model Training**

Besides *Doc2vec* and the logistic regression, all of our models are optimized using the *Adam optimizer* with early stopping, which stops the training if the improvement in the loss is less than $1e-4$ for three consecutive epochs. For all the experiments, we set the initial learning rate to 0.001, and set the weight decay to $1e-4$. Our main implementation is with Tensorflow, and all the computations are conducted on a 16-core Linux cluster with 128 Gb memory, and two Nvidia Tesla V100 GPU each with 16 Gb memory. We use the Doc2vec subroutine from the Gensim package[7] to pretrain the movie embeddings for ML-1m task2.

**Train/test split and metrics**

Since the goal of our experiments is not to find the best modelling and training configuration, we do not use a validation set to tune the hyperparameters. Instead, we provide sensitivity analysis on certain parameters of interest in Appendix F.3.

For **ML-1m task1**, we randomly split the movies by 80%-20% to construct the training and testing set for genre classification. For evaluation, we use the *accuracy* and *Macro F1 score* as metrics.

For **ML-1m task2**, we follow the convention of using the last user-movie interaction for testing, and use all the previous interactions for training. For evaluation, we use *Recall@5*, i.e. if the movie that the user truly viewed is among the top-5 recommendation, and *NDCG@5* that further discounts the position of the viewed movie in the top-5 recommendation.

---

[7]https://radimrehurek.com/gensim/models/doc2vec.html

## F.3 SUPPLEMENTARY RESULTS

We provide the sensitivity analysis for the featurization method. We focus on two variables, the dimension $d$ and the variance of $Q$ (denoted by $\sigma_Q^2$). Recall that we consider $Q$ as Gaussian distributions. We vary $d$ in $\{16, 32, 64\}$, and vary $\sigma_Q^2$ in $\{0.2, 0.4, 0.6, 0.8\}$.

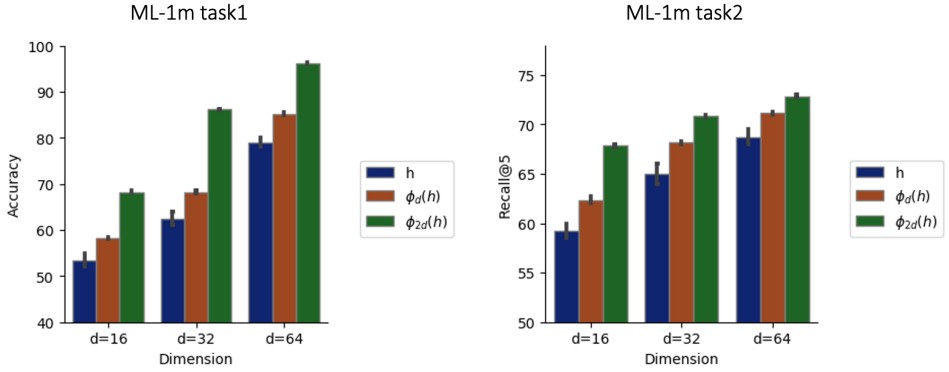

Figure A.3: Sensitivity analysis on the representation's dimension.

In particular, we first compare side-to-side $h_d$, $\phi_d(h)$, and $\phi_{2d}(h)$, while fixing $Q$ as the standard Gaussian distribution. We see from Figure A.3 that $\phi_d(h)$ consistently outperforms $h_d$ on both *ML-1m task1* and *ML-1m task2*. $\phi_{2d}(h)$ also significantly improves upon the performance of $\phi_d(h)$, which suggests the benefits of allowing extra model complexity in the particular tasks we consider. Further, the performance of both $\phi_d(h)$ and $\phi_{2d}(h)$ have considerable smaller variances than $h(x)$.

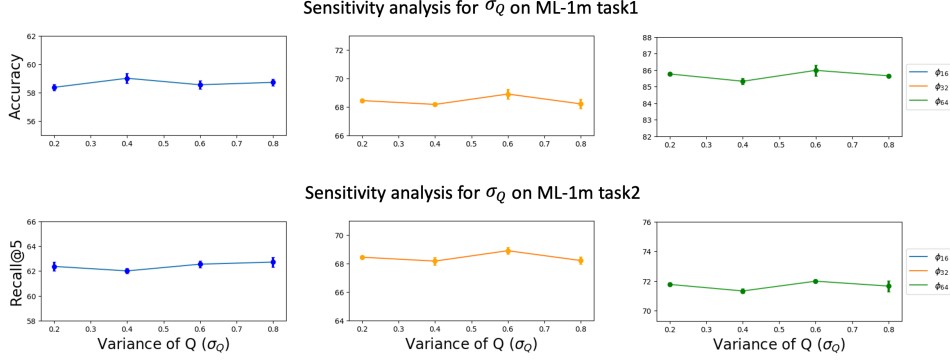

Figure A.4: Sensitivity analysis on the variance of $Q$ ($\sigma_Q$) from which $U$ is drawn.

We then examine the sensitivity of the downstream performance w.r.t. $Q$ – the sampling distribution for constructing $\phi_d(h)$. As stated before, we let $Q$ be zero-mean Gaussian distribution, and vary its variance. From Figure A.4, we observe that for all the dimensions we consider, the downstream task under $\phi_d(h)$ is very stable under different $\sigma_Q$. This echos Corollary 4 that our approach enjoys robustness to the selection of $Q$. In real-world productions, we have been using standard Gaussian distribution and observed no issues.

## F.4 ONLINE DEPLOYMENT

To avoid potential conflict of interest, we provide an overview of our production experiments. We aim to provide enough detail for interested practitioners to draw inspiration for both developing their own solutions and replicating ours.

**Some background introduction**.

In e-commerce application, the representation of items serves as a central component for almost all machine learning algorithms Wang et al. (2018); Xu et al. (2021). In the past few years, we have built a dedicated item representation learning pipeline that uses multiple sources of data to optimize item embeddings. Since there are **over billions of items** on our platform **Ecom**, it took us considerable effort to optimize the data pipeline and training routines such that the model refresh can be done on a daily basis.

We point out that the daily refresh is necessary for item representation learning because the catalog of items, which is a major source of pretraining data, also gets minor updates on a daily basis. For example, new items can be added, and the features of items (e.g. title, price, description) can be modified by the vendors. The other major source of pretraining data is the historical customer behavior data. They are critical for revealing the relationship (e.g. *similarity, complementariness, compatibility, substitutability*) among items. These relationships are relatively stable in the customer population, so the more data we use, the more likely for us to discovery useful signals.

Our model for pretraining item embeddings has both feed-forward component, recurrent-unit component, as well as contrastive learning component. The reason for using these different components is to effective handle data that has different structures. It is expected that the pretrained item embeddings are **stable**. As we mentioned above, the relationship among items are quite stable, and the catalog data has very minor differences in a limited span of time. Therefore, downstream machine learning models may follow a weekly or bi-weekly refresh schedule and are expecting very stable performances.

The four major applications that depend on our pretrained item embeddings, which we first introduced in Section 6, are ***item-page recommendation***, ***search ranking***, ***email recommendation***, and ***home-page marketing***. Each of the four tasks use both item embeddings and task-specific features to optimize their objectives. Most of them use model structures similar to the ***Wide and Deep*** network (Covington et al., 2016) to effectively combine information from different sources. Item-page recommendation aims to provide items that are related to the anchor item on that particular page that the customer is viewing. Item embeddings are in both the *recall* and *reranking* stage. Search ranking is a huge system that combines multiple components. In particular, the item embeddings are used in a particular *recall* stage. Email recommendation is a simpler task that aims to recommendation items related to what the customers recently interacted with, or are supposed to interact again. Item embeddings is used along with other features to build a model that optimizes CTR. Marketing is also a huge system in Ecom, and the particular component that uses item embedding is to build the *predicted click-through rate* model to support *bidding* and *placement ranking*.

**Brief summary of the production environment and implementation**.

Our production environment is relative standard in the e-commerce industry, with *Hive/Spark* supporting the offline data streaming and *Tensorflow Server* supporting the online inference of deep learning models.

Featuring $h(\cdot)$ via $\phi(h(\cdot), Q)$ can be easily implemented in production. Some practical advantages are:

- the algorithm is very simple and requires no training;
- it fits seamlessly into the current big-data infrastructure and frontend service;
- it require no change to the downstream model;
- the overhand for both the training and inference time are small;
- the communication can be easily done by simply recording the dimension and random seed under which a particular $U$ is generated.

On the **backend**, featurizing pretrained representation is engineered into a subroutine (on top of the original automated representation learning pipeline) callable by downstream applications. For instance, it can be a simple *PySpark* function if the end point of the automated representation learning pipeline is a *feature store* in *HDFS*.

The dimension $m$ and the random seed for generated the random directions $U = [u_1, \ldots, u_d]$ are the two main inputs. *Configuring and logging* the random seed used by each experiment is important because $U$ might be reused for deploying the model on frontend. If the dimension and random seed

are logged in the configuration, then there is no need to pass the particular $U$ around across different infrastructures. We mentioned in Section 5 that the dimension can be chosen by either specifying a threshold on the approximation error $\epsilon$ (Proposition 4), for which we provide an implementation that takes $\epsilon$ as input. We find that in most cases, downstream model owners are more willing to treated as a tuning parameter.

For **frontend** service, notice that featurizing $h$ into $\phi_m(h, Q)$ amounts to simply adding an initial fully-connected layer whose parameters are given by the $U \in \mathbb{R}^{d \times m}$ used in offline training. The activation function of that initial layer is given by the $\sin(\cdot)$ and $\cos(\cdot)$ functions. Therefore, it fits seamlessly with the *serialization* or *signature building processes* of the downstream model at no extra complication. In term of the inference time, it costs little overhead for large models. For smaller models and applications, on the other hand, we find out that it might be more efficient to directly *cache* the featurized $\phi_m(h, Q)$ at the expense of space complexity.

**A/B testing and result analysis**

Deploying a unified A/B testing across the different downstream applications is extremely challenging. Therefore, the A/B testing is conducted with each downstream application, but we coordinated with the different teams such that they approximately start at the same time.

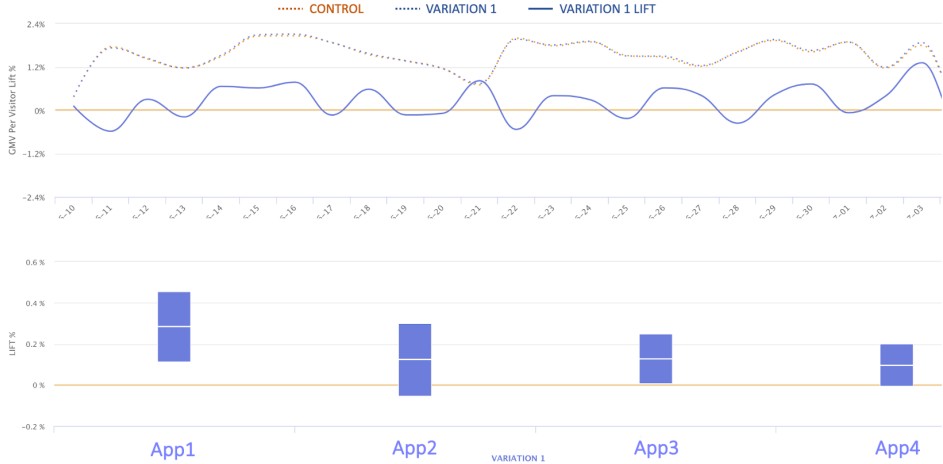

Figure A.5: Online A/B testing results, and the monitoring the per-vistor gross-merchandise value (**GMV**) during the testing period. **App1**, **App2**, **App3**, **App4** correspond to *item-page recommendation*, *search ranking*, *email recommendation*, and *home-page marketing*.

The **control baseline** is simply the existing implementation of each downstream applications. They use the raw item embeddings $h$ to refresh their models as usual. The **treatment** is switching to use $\phi(h)$ in their implementations and obtain another version of their models. The outputs of the control model and treatment model are exposed to users pre-allocated into different buckets. The exact testing logic is quite involved and beyond the scope of our paper. The key takeaway is that the testing begins almost at the same time for all four applications, and the only treatment variable is which item embedding to use. The detailed testing results are provided in Figure A.5. For brevity, we only show the performance in terms of CTR since it is indicative for all four applications.

As we shown in Table 1, all four A/B testings achieves statistically significant result under the level of $0.1$. In particular, the lift of item-page recommendation CTR has p-value less than $0.01$. Furthermore, we observe a very positive change to the per-visitor gross-merchandise value (**GMV**) during the testing period. Although this metric can be confounded by many other factors, we nevertheless believe featurizing pretrained representations benefits both the individual machine learning task and the overall business metric.

**Summary**

We examine the performance of featurizing pretrain item embedding via the method proposed in this paper. We discuss how the approach fits seamlessly with our production scenarios, the overview of our implementation, as well as the final testing results. Based on our experience, featurizing

representation could be an important step toward more stable and productive system for industrial machine learning.

## G  ADDITIONAL DISCUSSIONS

In this part of the appendix, we will provide additional discussions that provide readers with better understandings of our contributions.

**Connections between our theoretical and empirical results**

Proposition 1 and its subsequent discussion outline a potential problem with using pretrained representations in practice: the variance of $h(x)$ may cause the empirical downstream solution not concentrating to the optimal downstream solution. In Proposition 2, we provide a comprehensive result that characterize the variance of $h(x)$, which corroborates our empirical findings in Figure 1. In particular, it shows that although the pretraining algorithm may converge, the entires of $h(x)$ can remain stochastic, which negatively affects the downstream performance. Together, Proposition 1 and 2 suggests that directly using $h(x)$ in downstream tasks may result in poorer performances, which is corroborated by the empirical results in Table 1.

Proposition 3 and Lemma 1 analyze the potential interactions scenarios between pretrained representation and downstream tasks. They highlight the role of the embedding kernel and motivate the featurization approach we later developed. Proposition 4 provides stability guarantee for the proposed featurization approach, which indeed delivers superior performances as shown in Table 1.

**Additional relevant literature**

Besides the recent develop in machine learning theory summarized in Section 2, our work is related to a number of active domains in machine learning.

In terms of applications, pretrained representations are being used by a broad range of industrial machine learning tasks including information retrieval, recommeder systems, advertisement, knowledge completion, social network, user understanding, and many others (Cheng et al., 2016; Fan et al., 2022; Xu et al., 2020a; Zhang et al., 2018). Unfortunately, no existing work investigates their effectiveness from the industry's standpoint, that is, whether they strike a good balance between stability, predictability, and computability. Parallel to the above industrial tasks are the large natural language and computer vision models (Devlin et al., 2018; He et al., 2019), whose complexity exceeds the typical settings studied in this paper.

The proposed featurization approach relies on random Fourier feature – a well-established and important tool of modern machine learning (Rahimi & Recht, 2007). Its properties are known and are being actively applied to obtain kernel representations for many applications, such as signal processing, time series, and graphs (Liu et al., 2021; Gogineni et al., 2020; Xu et al., 2020b).

Finally, compared with the recent line of research on interpretability of deep learning models (Chakraborty et al., 2017), we focus particularly on the properties of learnt representations and how to improve them.

**Experiments with NLP dataset**

We further conduct experiments with the IMDB dataset described in Appendix F, where we use Word2vec for pretraining word embeddings, and apply them to a downstream bi-directional RNN model for sentiment analysis. We mention that large language models are not suitable for our purpose because the existing model checkpoints provided by the developers are from one-off implementation. They cannot be used to recover the variance information of the pretrained embeddings. Also, our analysis requires retraining those large models from scratch multiple times, which are very time-consuming and resource-intensive.

The experiments are straightforward, they closely reveal the instability issues of pretraining embedding and the benefits of the proposed featurization approach. In particular, we consider the window size (#ws) and number of negative samples (#ns) as the configuration parameters of the Word2vec model. We vary their values and present the array of downstream performances. For each pretraining configuration, we conduct 10 independent runs to compute the empirical variance as a measure of $h(x)$'s instability in the downstream task. Our Word2vec pretraining follows

the original implementation of (Mikolov et al., 2013), which uses the standard stochastic gradient descent with a decaying learning-rate schedule for 30 epochs. All the remaining settings are kept as default, and we observe they are sufficient for achieving very small training loss ($< 1e\text{-}6$).

The architecture of the downstream sentiment classification model is shown in Figure A.6, which is a bidirectional RNN standard in the NLP literature. To align with the original Tensorflow implementation[8], we adopt the same model configuration and treat the weight decay $\lambda$ as the only tuning parameter of the downstream task. For each set of pretrained word embeddings, we select $\lambda$ from {1e-6, 1e-5, 1e-4,1e-3, 1e-2} according to the validation accuracy.

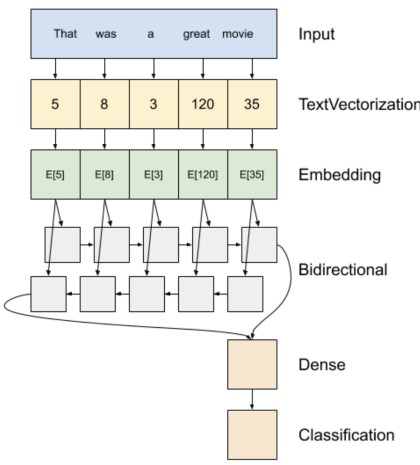

The train-validation-test split follows the standard 80%-10%-10%. We use the binary cross entropy loss and use accuracy as the metric. We also use the Adam optimizer with initial learning rate as 1e-4 and early stopping (when the validation accuracy stops improving for five epochs). The remaining settings are kept as default.

For reference, when the embedding layer is trained jointly, the downstream accuracy following the above implementation is: **86.25 (0.12)**. The downstream accuracy of using the different sets pretrained embeddings are provided in Table A.1. The featurized representations $\phi_d(h)$ is obtained in the same way as described in Appendix F. In Table A.1, we observe almost identical patterns to our previous results in Table 1. In particular, it shows that directly using $h(x)$ in the downstream task results in a high degree of instability under all pretraining configurations. The proposed featurization approach not only achieves better testing accuracy, but it improves the downstream stability by almost eight times on average.

Figure A.6: The model architecture of downstream sentiment analysis[8]. In the original experiments, the embedding layer is trained jointly with the rest of the model. Here we replace it by pretrained word embeddings.

Results on the NLP task provides additional evidence for both the instability issue of pretrained representations, and the benefits of the proposed featurization approach.

| d=16 | #ws=3, #ns=3 | #ws=3, #ns=4 | #ws=4, #ns=3 | #ws=4, #ns=4 |
|---|---|---|---|---|
| $h(x)$ | 74.87 (1.98) | 75.52 (1.96) | 76.02 (1.96) | 75.79 (1.97) |
| $\phi_d(h(x))$ | **76.65 (.24)** | **77.34 (.22)** | **77.95 (.22)** | **77.50 (.23)** |
| d=32 | #ws=3, #ns=3 | #ws=3, #ns=4 | #ws=4, #ns=3 | #ws=4, #ns=4 |
| $h(x)$ | 76.64 (2.01) | 78.23 (1.98) | 79.33 (1.75) | 78.87 (1.82) |
| $\phi_d(h(x))$ | **78.26 (.23)** | **80.07 (.21)** | **81.05 (.20)** | **80.79 (.21)** |
| d=64 | #ws=3, #ns=3 | #ws=3, #ns=4 | #ws=4, #ns=3 | #ws=4, #ns=4 |
| $h(x)$ | 76.35 (2.14) | 78.31 (2.16) | 78.48 (2.14) | 78.22 (2.09) |
| $\phi_d(h(x))$ | **78.49 (.24)** | **80.56 (.24)** | **80.86 (.21)** | **80.32 (.22)** |

Table A.1: Testing accuracy of using pretrained word embeddings for downstream sentiment classification. All results are multipled by 100, and in the paranthesis is the standard deviation computed from 10 independent pretraining runs.

---

[8] https://www.tensorflow.org/text/tutorials/text_classification_rnn

