# OpenReview forum: "Some Practical Concerns and Solutions for Using Pretrained Representation in Industrial Systems"
_ICLR.cc/2023/Conference — Submitted to ICLR 2023_

### Official Review · Reviewer_BWrq · 2022-10-22

**Confidence:** 3
**Clarity, Quality, Novelty And Reproducibility:** Have met the standards.
**Correctness:** 3
**Technical Novelty And Significance:** 3
**Empirical Novelty And Significance:** 3
**Recommendation:** 6

**Strength And Weaknesses:**

Pros:

-The problem addressed has high practical value: it tries to make pre-trained model more accessible to a range of industrial systems. The "Featurizing Pretrained Representations" idea will promote the performance of downstream tasks compared to other approaches.

-This paper verifies its thought with a detailed formula. By analyzing the features of the pre-training models, the author proves that the features from the pre-training models are related to the stability of the performance of the downstream tasks. In detail, the variance of the pre-training features h (x) plays an essential role in the learning process of downstream tasks, and the variance will increase with the increase of random variables in the pre-training model.

-The proposed method (Featurizing Pretrained Representations) is intuitive and effective, which uses a more stable h (x), e.g., the representation of weights in the NW estimator in Section 4.3, in downstream tasks. Moreover, experiments on six downstream tasks indicate that the method has superior performance and stability.


Cons:

-It may have a more suitable h(x). The authors think the optimal h(x) can be obtained through Fourier Transform. But it is not easy to get the representation of the sample by the inverse Fourier transform. So they replaced this method by increasing the number of samples and the dimension of features. Nevertheless, this method may not be the most suitable one in industrial systems due to the limited sample and computing resources. Maybe the authors can seek a better method from the perspective of math or Fourier Transform.
-The analysis is not comprehensive enough. Although the paper proves that the features of pre-training model do affect the stability of downstream tasks, it ignores the features of downstream tasks, which are more essential for downstream tasks. Have the authors tried to analyze the downstream tasks, and what does the result look like?

-The essence of h(x) is still not clear. The paper discussed the impact of the variance of h(x) on downstream tasks, and then the authors improved the performance with the optimization of h(x). However, they did not give any description of the reason why the pretraining models generate such features h(x). Maybe it is more important.


**Summary Of The Paper:**

The paper aims to solve the suboptimal problems caused by the pretraining models in industrial systems. And the authors analyze the problem with four aspects( the gap of uniform convergence for analyzing pretrained representations, their stochastic nature under gradient descent optimization, what model convergence means to them, and how they might interact with downstream tasks). Then they propose a method to refine the features of pretraining models in various ways to improve the stability of the downstream task model. Experiments on six tasks prove the effectiveness of the proposed method.



**Summary Of The Review:**

Overall, the paper proposed an interesting idea and showed strong empirical results, but it should have a more comprehensive analysis of both pretraining and finetuning. Hence, I would suggest the author complete more related works to give more convincing conclusions.

---

> ### Author Response · Authors · 2022-11-19
> **We thank the reviewer for the comments, and we find the reviewer's concerns are mostly caused by some misreadings and unfamiliarity with random Fourier features**
>
> We thank the reviewers for providing feedback on our manuscript, acknowledging our contributions, and suggesting possible areas for improvement.  Nevertheless, we believe the key concerns raised by the reviewer are caused by some misreadings and unfamiliarity with random Fourier features. We will make them clear for the reviewer in this rebuttal. Once these concerns are addressed, we believe reviewers' understanding of our work will improve significantly.
>
>
> **"It may have a more suitable h(x)"**
>
> It appears the reviewer is not very familiar with random Fourier features [1], which we fully understand and will explain it here. Briefly put, the idea of random Fourier feature is to use Monte-Carlo approximation to compute the inverse Fourier transform of certain kernel functions. In the machine learning literature, random Fourier feature is a well-established technique that has been widely used and studied, and its efficiency and optimality for representing kernels are also known [2]. Therefore, some of the reviewer's comments about our featurization approach are incorrect, including:
>
> --*"But it is not easy to get the representation of the sample by the inverse Fourier transform"*, whereas the sampling is actually simple and highly efficient, which is one of the key reasons why random Fourier feature is so popular;
>
> --*"So they replaced this method by increasing the number of samples and the dimension of features"*, whereas random Fourier feature has nothing to do with the number of samples because it only aims to represent the kernel, and the dimension is a choice of deign where larger is not necessarily better (think about the bias and variance tradeoff).
>
> Considering the paper on random Fourier feature has just won the test-of-time award in last year's NeurIPS, we believe the reviewer's comments that *"Maybe the authors can seek a better method from the perspective of math or Fourier Transform"* is certainly not appropriate given how successful random Fourier features are.
>
> While inventing a new random feature technique is out of the scope of this paper, our unique contribution on random Fourier features is to prove Proposition 4, which establishes the stability guarantee even if the correct spectral distribution is not used in practice. This result is novel and adds value to the existing framework of random Fourier feature, the interest of which may extend beyond this paper.
>
>
> **"The analysis is not comprehensive enough because downstream features are not included"**
>
> We agree with the reviewer that downstream features are indeed important for industrial applications, but **they rarely affect how pretraining representations are obtained**, and are thus not relevant to our main investigations. Further, the most common source of instability for downstream features is **measurement error**, which is drastically different from the source of instability for pretrained embeddings studied in this paper. Nevertheless, our analysis for the NW estimator might provide an entry point for studying possible interactions between pretrained representations and downstream features, when they are used together to define the distance between samples. Still, the topic is too broad to be studied in a single paper.
>
>
> **"The essence of h(x) is still not clear"**
>
> As the reviewer mentioned, the contribution of our work is investigating the stability and predictability of h(x) for general pretraining model and procedure. We made it clear in the *Takeaway* paragraph of Section 4.2 that the stochasticity of h(x) is due to the random initialization and the random order by which data is fed to SGD.  Our conclusion applies to *any* h(x) obtained this way, regardless of how they are generated by the pretraining model. Therefore, the reviewer's question of "why the pretraining models generate such features h(x)", though relevant to other purposes, does not concern our main analysis.
>
>
> Finally, we add to the revised manuscript a dedicated section (in **Appendix G**) to address the reviewers' request, including adding more related-work discussion.
> We again thank the reviewer for the time and effort devoted to our manuscript. We understand that the reviewer has broad interests in related topics, but there is always the **tradeoff between depth and width**. Ouur work focuses on the depth of the problems and aims to provide grounded analysis that is lacking in existing literature. By address the reviewer's key concerns, we hope this rebuttal can improve the reviewer's understanding of our manuscript.
>
>
> [1] Rahimi A, Recht B. Random features for large-scale kernel machines. NeurIPS 2007
>
> [2] Liu F, Huang X, Chen Y, et al. Random features for kernel approximation: A survey on algorithms, theory, and beyond. IEEE TPAMI, 2021.

---

> > ### Comment · Reviewer_BWrq · 2022-11-21
> > **post-rebuttal reply**
> >
> > I thank the authors for providing detailed response.
> >
> > With the extended interpretation and material, I agree to raise the score to 6.

---

### Official Review · Reviewer_VPk5 · 2022-10-23

**Confidence:** 4
**Correctness:** 2
**Technical Novelty And Significance:** 2
**Empirical Novelty And Significance:** 2
**Recommendation:** 3

**Clarity, Quality, Novelty And Reproducibility:**

(clarity) I think the paper is well written. It would be better if details on the data is provided (e.g., the dataset size), as the stability issues may be related to the small number of samples.

(quality) If my concerns are addressed, the paper quality is good as the main claim is supported by both theories and empirical results.

(novelty) I think the claim could be novel if my concerns on the proof are addressed.


**Strength And Weaknesses:**

**Strengths**:
* Provide theoretical interpretations on empirical observations.
* A solution to address the claimed instability issue is simple.

**Weaknesses**:
* The main empirical observations (Figure 1) are made in weak setups
* Proposition 1 is not necessarily true.
* The experiment result is weak

(Weakness 1)

The paper first made empirical observations on MovieLens-1m along with the traditional Doc2vec model as a pretrained representation, along with logistic regression for a downstream task. Considering a huge advance on a large language model, this experiment setup does not represent the current state of representation learning. If this observation is made with IMDB (or any larger dataset) along with BERT (or RoBERTa), the story would be much stronger.

Moreover, the following statement needs to be more carefully explained:
“Since we use logistic regression as the downstream model, the fluctuation can only be caused by the instability of pretrained representations.”
Why logistic regression cannot be the source of the instability? In particular, do you mean the optimization for the logistic regression is done until it converges to the global optimum?


(Weakness 2)

Proposition 1 implies that the uniform convergence bound for pretrained representation has an irreducible term, contributing to the instability of the downstream task performance. However, I think Proposition 1 is not necessarily true. In particular, (A.1) decomposes the generalization bound, and the proof claims some terms are non-negative, which is not always true.
Consider the second term in (A.1), where the proof said “by definition, the second term is non-positive “. First, I think "non-positive" is a typo as if it is non-positive, it can cancel out the irreducible term, thus the claim does not make sense; so I assume that the term is "non-negative". Here, h* is the optimal representation w.r.t. the expected loss R(h, f) instead of the empirical loss, thus the second term can be negative. In other words, if $h^* = argmin_h \sum_i \ell(f \circ h(X_i), Y_i)$, the second term cannot be negative; but the if statement does not hold. Additionally, the non-negative argument on the last term is the same issue.

This proof and the affected arguments in the following sections need to be fixed and adjusted.


(Weakness 3)

I liked the experiments on real-world production experiments, which demonstrates how knowledge in academia is actually used in practice. But, to justify the efficacy of the proposed approach, it is always good to evaluate multiple public datasets, but currently only one is used.

For the ML-1m, I’m not sure if this is a widely acceptable dataset in representation learning; the main reason is that its input dimension looks not large enough (e.g., based on the paper, saying “each movie is provided with its title and genre, in the form of English words or sentences. There are 18 genres in total.”). Based on this limited information, classifying five ratings sounds very challenging (contrast to this, a human rates each movie based on richer information). I personally think this unusual property of the dataset may lead to the unstable results in Figure 1. Appendix F1 includes some results on IMDB, but I cannot see the downstream accuracy results as in the figure in page 2.

In short, evaluating the proposed approach at least 2 widely acceptable datasets would make the experiment results stronger. I guess IMDB's downstream accuracy results need to be presented to strongly show the observed instability issue. Moreover, I’d recommend using features from BERT or RoBERTa for pretrained model features. Based on Table 1, the featurting pretrained representation looks promising, and hoping that the same trend holds for other datasets and representations.


**Summary Of The Paper:**

The paper observed an instability issue of downstream prediction based on features from pretrained models (i.e., the accuracy of a downstream task is varing, which depends on the pretrained representation trained on the same data, model configuration, training setup, and stopping criteria). This empirical observation is theoretically analyzed via a novel uniform convergence bound for pretrained representations, and eventually proposes a solution, called featurizing pretrained representation, which is empirically justified to outperform a baseline (i.e., a downstream task with usual pretrained features) using the MovieLens1m dataset and an in-house real-world production dataset.


**Summary Of The Review:**

I like the way of this paper to attack the observed representation instability; but the paper does not follow the actively changing trends — large language models are widely used, but didn’t account for this line of work (including in the related work section). Moreover, as mentioned in the weaknesses, I have concerns on the proof that supports the paper’s main contribution. Thus, I vote for rejection for now, but I’m willing to adjust my understanding.


=== After the rebuttal

Thanks for providing details.

For the Proposition 1 proof, it needs to provide that the slack term does not disappear for some f and h. In particular, the generalization error in Proposition 1 is decomposed in three terms in the proof. And each term is *independently* bounded by a uniform convergence bound. However, this does not mean that there exist f and h such that the slack term is not canceled by other terms. Moreover, the argument on the existence of an irreducible slack term may not be achievable via upper-bounding the generalization error since I can take any upper bound and then artificially add some slack term to claim that there is an irreducible slack term.

I think having a correct proof Proposition 1 is necessary to generalize the paper’s intriguing observation — “In real-world production, however, we have encountered key problems that cannot be justified by existing knowledge. They raise concerns that the naive use of pretrained representation as feature vector could lead to unwarranted and suboptimal solution.” I agree that Proposition 1 is an intermediate step, but this is the basis of the paper’s argument. Otherwise, authors’ need to empirically justify their observations with a large number of experiments with various models and datasets.

For these reasons, I still lean toward rejection, but I would see the value of the proposed algorithms (as heuristic) and empirical results.

---

> ### Author Response · Authors · 2022-11-19
> **We thank the reviewer for the valuable feedback, but we are afraid the reviewer has some wrong conceptions about our work**
>
> We sincerely thank the reviewers for carefully reading our manuscript and suggesting areas for improvement, especially for going over our proof.
> However, we find that the reviewer has some incorrect conceptions about our proof and experiments, which we will explain in detail below. Also, it seems the reviewer has some misunderstanding about our experiments and unfamiliarity with the recommendation literature, which leads to some incorrect comments under Weakness 3. In this rebuttal, we will address the concerns and questions about the proof and experiments. We also add an additional set of experiments as suggested by the reviewer. By make clearing of the rationality behind our proof and experiments, we believe the reviewer's understanding of our contributions will improve significantly.
>
>
> **To Weakness 1**
>
> The reviewer raises a good question why we choose shallow embedding models to corroborate our analysis. The most important reason is that using complex models will inevitably introduce **confounding factors**, which can cause **gaps** for empirical analysis. These confounding factors include optimization methods, e.g. adding momentum, learning rate schedule, or other adaptive components, how to ensure the model confiugrations are tuned appropriately, and whether the pretraining finds the optimal solution. Justifying all those factors are highly non-trivial, as some of them are still open questions in the community. Using shallow models helps us avoid the gaps between the empirical and theoretical analysis, which is necessary for the rigor and straightforwardness of our analysis.
>
> We also kindly disagree with the reviewer on the comment *"Considering a huge advance on a large language model, this experiment setup does not represent the current state of representation learning"*. We understand that the reviewer has high expectations on the industry's adoption of cutting-edge ML technology, but the reality is that practitioners and engineers prefer solutions that are **safer, well-understood (less complex), and are tested by time**.
>
> Large language models, even if used, can only support a small proportion of industrial tasks. The majority of industry ML applications, including online e-commerce, advertisement, job matching, social network, ride sharing, grocery delivery, require pretrained representation of categorical variables (such as user, product, type of service, device, etc) that are not NLP-driven. In many industrial systems, shallow embedding models are used intensively due to their interpretability and computational efficiency. We encourage the reviewer to not judge our work by what fancy model we use, but the **depth, incisiveness, and practicality** of our analysis since they can apply to a broad range of problems.
>
> Finally, the reviewer correctly understands that we use logistic regression in downstream tasks because it can be efficiently optimized to the global optimum. Therefore, it does not introduce an additional source of instability. We add this explanation to the revised manuscript to avoid further confusions. We also add an additional set of experiments as suggested by the reviewer, which we will discuss in our response to Weakness 3.
>
>
>
> **To Weakness 2**
>
> We thank the reviewer for check our proof with care. However, the reviewer's comments are not correct. For the second term in (A.1):
> $\min_{f\in\mathcal{F}} \frac{1}{n} \sum_i \ell\big(f\circ \hat{h}(X_i), Y_i \big) - \min_{f\in\mathcal{F}} \frac{1}{n} \sum_i \ell\big(f\circ h^*(X_i), Y_i \big) ,$
> recall that that $\hat{h}$ is the empirical minimum, whereas $h^*$ is the expected minimum. Note that empirical minimum, when evaluated on the empirical distribution, is always no larger than the expected minimum when also evaluated on the empirical distribution. Since both terms are evaluated under the empirical expectation here, the empirical minimum (which is the first term) will lead to a strictly smaller or equal outcome than the expected minimum (which is the second term). So their difference is always non-positive. The reviewer's suspicion that $\min_{f\in\mathcal{F}} \frac{1}{n} \sum_i \ell\big(f\circ h^*(X_i), Y_i \big)$ can be negative is not wrong, but if this term is negative, the first term can be more negative so their difference is still non-positive.
> The same argument applies to the last term in (A.1).
>
> Therefore, the proof is **correct as-is**.
>
> **To Weakness 3**
>
> (see the continuation of this thread)

---

> > ### Author Response · Authors · 2022-11-19
> > **We thank the reviewer for the valuable feedback, but we are afraid the reviewer has some wrong conceptions about our work**
> >
> > **To Weakness 3**
> >
> > We understand that the reviewer might not be familiar with the recommendation literature, which causes some major misunderstandings of our experiments. We will explain them here for the reviewer.
> >
> > Firstly, recommendation and content understanding (e.g. user or item categorization) are two of the most important tasks in the industry, and Movielens-1M (ML-1M) is **the most popular dataset** for representation learning in recommendation and content (movie) understanding. It contains around one million records and is a decent large benchmark dataset. We kindly remind the reviewer that most representation learning datasets used by academia does not have the necessary elements of an industrial system including user-product interaction and their contextual data. ML-1M is one of the few options in this regard.
> >
> > Secondly, we believe the reviewer misunderstands how downstream recommendations are made. In short, we are doing **sequential recommendation** instead of traditional collaborative-filtering recommendation. We aim to predict which movie will be rated next, not the exact rating. Also, the prediction is based on the user's rating history, not the contextual features of the users and movies. Furthermore, the movie information is only used to pretrain the movie embedding, which serve as features in the downstream sequential recommendation model. In particular, the two-tower recommendation model in our experiments (Figure A.2) is very common in the recommendation literature.
> >
> > Therefore, the reviewer's comment *"... this unusual property of the dataset may lead to the unstable results in Figure 1 ..."* is incorrect, since everything in our experiments follows the standard practice in recommendation literature, and the unstable results are indeed caused by the variance of the pretrained representations.
> >
> > Finally, we add an additional set of experiments on the IMDB dataset as suggested by the reviewer.
> > In particular, we use Word2vec to pretrain word embeddings, and use a bi-directional RNN model for downstream sentiment classification. The details and results are added to **Appendix G** of the revised manuscript. In short, we observe very similar patterns to our existing results on this NLP task. We mention that the BERT-family model suggested by the reviewer are **not suitable for our purpose** because the existing model checkpoints provided by the developers are from one-off implementation. They cannot be used to recover the variance information of the pretrained embeddings. Also, our analysis requires retraining those large models from scratch multiple times, which are very time-consuming and resource-intensive.
> >
> > We again thank the reviewer for devoting time and effort to our manuscript, and we look forward to the future discussions.

---

> > > ### Comment · Reviewer_VPk5 · 2022-11-19
> > > **Thanks**
> > >
> > > I appreciate the authors’ response and additional experiments. The following includes remaining concerns.
> > >
> > > **Weakness2**. Here, I assume “non-positive” in “Therefore, the last term is always non-positive” is a typo. If it is not true, how can we know that this possibly negative term does not cancel out the slack term (that does not vanish as we increase n)?
> > >
> > > Based on what is written in the proof, the possibly negative second and third terms can cancel out the slack term, thus the value of Proposition 1 diminishes. This may need additional clarification.
> > >
> > >
> > > **Overall**. I think the value of this paper lies in the intriguing theoretical finding (assuming Proposition 1 is correct). In this sense, whether industry needs practical models may not matter too much. The initial observation may be obtained from industry-level models, but the paper claims that the intriguing property holds in general by proposing theories. At that point, as a reviewer, I need to see whether the theoretical findings can be justified from empirical results with various models and datasets. That’s why I think it is important to use currently widely used models (e.g., BERT) or well-known datasets along with these models.

---

> > > > ### Author Response · Authors · 2022-11-20
> > > > **Thank you for the updated comments, we address the questions and express our concerns about the evaluation**
> > > >
> > > > We sincerely thank the reviewer for responding to our rebuttal. The reviewer's new question on our bound is actually **against** the worst-case nature of **uniform convergence**, which we explain below.
> > > >
> > > > The uniform convergence bound is intended to hold for all hypothesis, which means it will require upper bound the supermum of the empirical processes (including upper bound by zero if strictly non-positive). The scenario mentioned by the reviewer *"how can we know that this possibly negative term does not cancel out the slack term"* does not work in the context of uniform convergence, because we can construct toy examples under which the last term is zero in the worst case (imagine n=1 and everything is convex). We mention that our arguments and procedure are all standard in the ML theory literature, and that the value of Proposition 1 is solid as a uniform convergence bound.
> > > >
> > > > On the other hand, we are concerned about the reviewer's opinion about our contributions. We understand that Proposition 1 has potential values and its interest may extend beyond the scope of our work. Adding more experiments will make it stronger. However, Proposition 1 only serves an intermediate step to illustrate the problem we investigate and address.
> > > > Not to mention that its implications have already been elaborated by the existing experiments in our manuscript (including the new experiment suggested by the reviewer).
> > > > What confuses us is that the reviewer deems not having more sophisticated experiments for an intermediate illustrative result is a major flaw of our work.
> > > > While we respect the reviewer's judgement and trust ICLR's reviewing guidelines, we are concerned that this is **not a fair evaluation of our work as a whole**.
> > > >
> > > > We appreciate the reviewer for showing high standards and professionalism in the discussion, and we believe the reviewer will take our concern into consideration. We look forward to future discussions.

---

> > > > > ### Comment · Reviewer_VPk5 · 2022-11-28
> > > > > **Thanks**
> > > > >
> > > > > Thanks for providing details. Here, I highlight my understanding on Proposition 1, but see the after-rebuttal comment for details.
> > > > >
> > > > > For the Proposition 1 proof, it needs to provide that the slack term does not disappear for some f and h. In particular, the generalization error in Proposition 1 is decomposed in three terms in the proof. And each term is *independently* bounded by a uniform convergence bound. However, this does not mean that there exist f and h such that the slack term is not canceled by other terms. Moreover, the argument on the existence of an irreducible slack term may not be achievable via upper-bounding the generalization error since I can take any upper bound and then artificially add some slack term to claim that there is an irreducible slack term.

---

> > > > > > ### Author Response · Authors · 2022-12-05
> > > > > > **We thank the reviewer for the clarification, and add minor modifications to our proof to address the reviewer's concern**
> > > > > >
> > > > > > We appreciate the reviewer for further clarifying the question and continue the discussion with us. We now have a better sense of your reasoning process, and would like to propose minor modifications to our proof to meet the reviewer's demand.
> > > > > > We understand the reviewer's concern that there might be terms cancelling out the non-concentrating slack term, which makes our bound loose and thus not meaningful. Here, we show that the reviewer's concern can be effectively eliminated, and elaborate the technical assumptions needed.
> > > > > >
> > > > > > *(1)* Only the first two terms in (A.1) contain $\hat{h}$ (over which we will be taking the supremum), and the non-concentrating slack term is a consequence of applying Lemma A.2 to the first term. Lemma A.2 is known to be tight because it is based on the Mcdiarmid's inequality, so there should be no concerns with it. Assuming that $\mathcal{H}$ contains $h^*$ (which is a standard and mild assumption).
> > > > > > Then notice that the negative non-concentrating slack term cannot arise from the second term because obviously the second term equals to zero when $\hat{h} = h^*$. When this happens, the first bound (which should hold for any $h \in\mathcal{H}$) still has the non-concentrating slack term as Lemma A.2 is tight.
> > > > > >
> > > > > > *(2)* Neither the third or forth term contains $\hat{h}$, which makes our following argument more straightforward. Notice that the negative non-concentrating slack term *cannot arise from the third term* because it is of $\mathcal{O}(1/\sqrt{n})$ if the second moment of each individual loss term is of $o(1)$, which is also a standard and mild assumption people use to achieve sharper bounds. This can be seen by simply applying the *Bernstein bound* instead of the Hoeffding bound which we used previously. Finally, the negative non-concentrating slack term *cannot arise from the forth term* as well, because the forth term is simply a reversed excessive risk term which also has a rate of $\mathcal{O}(1/\sqrt{n})$ as a standard result.
> > > > > >
> > > > > > Therefore, we only need to **add two standard assumptions**, i.e. the second-moment assumption on the individual loss term and that $h^*\in\mathcal{H}$, to show that the non-concentrating slack term cannot be potentially eliminated. This should address the reviewer's final concern once and for all, and we hope our proposed modifications can be reflected in the reviewer's final evaluation. We will add these modifications to the next revision of our manuscript, and we again thank the reviewer for challenging our technical details and helping us improve our work.

---

### Official Review · Reviewer_vPVs · 2022-10-29

**Confidence:** 2
**Clarity, Quality, Novelty And Reproducibility:** Paper is clearly written and the cont…
**Correctness:** 3
**Technical Novelty And Significance:** 3
**Empirical Novelty And Significance:** 3
**Recommendation:** 6

**Strength And Weaknesses:**

Strengths:
1) Authors explain concerns with existing methods clearly.
2) Authors divide each of their theoretical results and give a takeaway summary for better understanding for readers
3) Authors back their theoretical claims with empirical results

Weaknesses:
1) Setting considered in the paper is too simplistic to be broadly used in the industry.
2) More connection between theoretical results and empirical results is not discussed well enough.


**Summary Of The Paper:**

The Author's investigation reveals critical insights into the gap of uniform convergence for analyzing pre-trained representations, their stochastic nature under gradient descent optimization, what model convergence means to them, and how they might interact with downstream tasks. Authors propose a simple approach which contributes to both applied and theoretical research of representation learning.

**Summary Of The Review:**


1) In section 3, why is \Theta d x k when y is just a scalar? How to make sense of loss in this case?

2) Section 4.2 "instability of ˆh(x): the exact position of ˆh(x) in Rd is stochastic, depending on the initialization and the order of the pretraining data that is fed to SGD": What if the initializations are same every time?

3) "In addition to improved stability and performance": How is stability measured in Table 1?

4) How was hyperparameter tuning done for the main results?

---

> ### Author Response · Authors · 2022-11-19
> **We thank the reviewer for carefully reviewing and commenting our manuscript, and we address the key questions and concerns in the rebuttal**
>
> We thank the reviewer for acknowledging the contribution of our work and for pointing out weaknesses and potential areas for improvement. We agree that real-world applications can be more complex than the settings we study in this paper, however, they are less amenable to in-depth investigation (which is the main goal of our paper). Furthermore, the results and insights in our work can be generalized to a broad range of settings.
> The compromises we made are necessary to achieve a clear, thorough, and rigorous analysis of the problem.
> They would not have been possible if we had started with a very complex setup.
>
> To further complement the existing experiments in our manuscript, we conduct additional experiments on a NLP tasks using language models.  The detailed setup and results are added to **Appendix G** of the revised manuscript. In short, we observe very similar patterns in the new experiments, i.e. both the instability issue of pretrained representation and the benefits of the proposed featurization approach. The new results provide additional empirical evidence that supports our key contributions in a more complex setting.
>
> We address the reviewer's main concerns as follows.
>
> **Setting considered in this paper is too simple to be broadly used in the industry**
>
> We understand that the reviewer has high expectations on the industry's adoption of cutting-edge ML technology, but the reality is that practitioners and engineers prefer solutions that are **safer, well-understood (less complex), and are tested by time**. Unlike the academia where novelty and complexity are important criteria, in industry, there are numerous business, data, and infrastructural infrastructure constraints that require solutions to be stable, predictable, and computable -- these are the three pillars of our work. Therefore, we encourage the reviewer to not judge our work by the complexity of the setting but the **depth, incisiveness, and practicality** of our theoretical and empirical investigations.
>
> **More connections between theoretical and empirical results is not discussed well enough**
>
> In Appendix G of the revised manuscript, we add a dedicated paragraph to further explain the connections as suggested by the reviewer, which is given below.
>
> Proposition 1 and its subsequent discussion outline a potential problem with using pretrained representations in practice: the variance of h(x) may cause the empirical downstream solution not concentrating to the optimal downstream solution. In Proposition 2, we provide a comprehensive result that characterize the variance of h(x), which corroborates our empirical findings in Figure 1. In particular, it shows that although the pretraining algorithm may converge, the entires of h(x) can remain stochastic, which negatively affects the downstream performance. Together, Proposition 1 and 2 suggests that directly using h(x) in downstream tasks may result in poorer performances, which is corroborated by the empirical results in Table 1.
>
> Proposition 3 and Lemma 1 analyze the potential interactions scenarios between pretrained representation and downstream tasks. They highlight the role of the embedding kernel and motivate the featurization approach we later developed. Proposition 4 provides stability guarantee for the proposed featurization approach, which indeed delivers superior performances as shown in Table 1.
>
> We answer the remaining minor questions as below.
>
> **Q1** -- In this question, the reviewer refers to the multi-class scenario (where k is the number of classes). In this case, the loss will be given by such as the categorical cross-entropy loss, which will not affect the remaining arguments in our paper.
>
> **Q2** -- Importantly, random initializations are critical to the success of using SGD on non-convex landscapes, so it will always contribute the variance of h(x). Therefore, it is not meaningful to condition the analysis on a specific realization of the random initialization. Even if the intializations are fixed every time, the random order by which samples are fed to SGD will still cause non-trivial randomness to h(x).
>
> **Q3** -- The stability in Table 1 is measured by the standard deviation of results obtained from ten independent pretraining runs. Each independent run differs only in the random initialization and the order of samples in SGD optimization. Since we are using logistic regression in the downstream task, there is no need to repeat the downstream training since the global optimum solution can be effectively obtained.
>
> **Q4** -- An important reason for using reduced settings is that there are very few confounding factors that can affect the interpretation of results. In most of our benchmark experiments, the main configurable parameters are either set to fixed values or examined by sensitivity analysis (the details are provided in Appendix F).
>
> We again thank the reviewers for their time and effort with our manuscript, and we look forward to future discussions.

---

> > ### Comment · Reviewer_vPVs · 2022-12-07
> > **Response to authors**
> >
> > Thank you for clarifying some of my questions. After reading the response and other reviews, I would like to keep my current score (because of concerns on Preposition 1 as noted by other reviewer and lack of clear connection between theory and practice)

---

### Decision · Program_Chairs · 2023-01-20

**Decision:**

Reject

**Justification For Why Not Higher Score:**

The paper's writing could be improved. There's also not as strong a connection between empirical practice and theory due to a few simplified settings that they conducted for validation (also raised by multiple reviewers, e.g., the focus on Doc2Vec).

**Justification For Why Not Lower Score:**

N/A

**Metareview: Summary, Strengths And Weaknesses:**

This work investigates the challenges that face using pretrained representations in industrial systems. This is an important topic given the shift in deep learning toward leveraging large pretrained models. The authors provide theoretical analysis on the gap of uniform convergence for analyzing pretrained representations, as well as other analysis such as how this interacts with downstream tasks.

After discussion, reviewers recommended reject with the exception of 1 reviewer who did not participate in the discussions. The major concern is around the proof behind Proposition 1, which provides a uniform convergence bound arguing effectively for a "no free lunch” result for using pretrained representations. Reviewers agreed that Proposition 1 is crucial to the work as the rest of the analysis is based on it, and the additional assumptions the authors added after rebuttal still don't resolve the issue. The manuscript is also not updated to reflect any such changes. On the empirical side, one also needs to have variety of datasets and models to back their claims, but their empirical analysis is restricted to one pretrained model's embeddings and one dataset.